# Postsynaptic RIM1 modulates synaptic function by facilitating membrane delivery of recycling NMDARs in hippocampal neurons

Jiejie Wang[1], Xinyou Lv[1], Yu Wu[1], Tao Xu[1], Mingfei Jiao[1], Risheng Yang[2], Xia Li[2], Ming Chen[1], Yinggang Yan[1], Changwan Chen[1], Weifan Dong[1], Wei Yang[1], Min Zhuo[3,4], Tao Chen [2,3], Jianhong Luo[1] & Shuang Qiu[1,5]

NMDA receptors (NMDARs) are crucial for excitatory synaptic transmission and synaptic plasticity. The number and subunit composition of synaptic NMDARs are tightly controlled by neuronal activity and sensory experience, but the molecular mechanism mediating NMDAR trafficking remains poorly understood. Here, we report that RIM1, with a well-established role in presynaptic vesicle release, also localizes postsynaptically in the mouse hippocampus. Postsynaptic RIM1 in hippocampal CA1 region is required for basal NMDAR-, but not AMPA receptor (AMPAR)-, mediated synaptic responses, and contributes to synaptic plasticity and hippocampus-dependent memory. Moreover, RIM1 levels in hippocampal neurons influence both the constitutive and regulated NMDAR trafficking, without affecting constitutive AMPAR trafficking. We further demonstrate that RIM1 binds to Rab11 via its N terminus, and knockdown of RIM1 impairs membrane insertion of Rab11-positive recycling endosomes containing NMDARs. Together, these results identify a RIM1-dependent mechanism critical for modulating synaptic function by facilitating membrane delivery of recycling NMDARs.

[1] Department of Neurobiology, Institute of Neuroscience, NHC and CAMS Key Laboratory of Medical Neurobiology, Zhejiang University School of Medicine, Hangzhou, 310058 Zhejiang, China. [2] Department of Anatomy, Histology and Embryology, Fourth Military Medical University, 710032 Xi'an, China. [3] Center for Neuron and Disease, Frontier Institutes of Life Science, Science and Technology, Xi'an Jiaotong University, 710049 Xi'an, China. [4] Department of Physiology, Faculty of Medicine, University of Toronto, 1 King's College Circle, Toronto, ON M5S 1A8, Canada. [5] Department of Anesthesiology, Second Affiliated Hospital, Zhejiang University School of Medicine, Hangzhou, Zhejiang 310058, China. These authors contributed equally: Jiejie Wang, Xinyou Lv. Correspondence and requests for materials should be addressed to T.C. (email: chtkkl@fmmu.edu.cn) or to J.L. (email: luojianhong@zju.edu.cn) or to S.Q. (email: qiushly@zju.edu.cn)

N-methyl-D-aspartate receptors (NMDARs) and α-amino-3-hydroxy-5-methyl-4-isoxazolepropionic acid receptors (AMPARs) are major types of glutamate receptors that are widely distributed in the brain and play pivotal roles in synaptic function[1, 2]. AMPARs mediate most of the basal synaptic transmission, while NMDARs are important for triggering plastic changes. NMDAR activation initiates different signals that lead to rapid insertion of AMPARs into or internalization from the synapse, which mediate long-term potentiation (LTP) or long-term depression (LTD), respectively[3–6]. It is now accepted that NMDARs are not static, but undergo constitutive cycling into and out of the postsynaptic membrane and lateral diffusion between synaptic and extrasynaptic receptor pools[7–10]. Internalized NMDARs may be delivered to the late endosome, and then to the lysosome for degradation, or may be sorted to the recycling endosome for reinsertion to the plasma membrane[11, 12]. Furthermore, the number and subunit composition of synaptic NMDARs are dynamically regulated during development- and experience-dependent neuronal activity[8, 13]. NMDAR-mediated LTP or LTD responses have been induced at different synapses by different patterns of synaptic activity[14–18]. In some pathological conditions, such as chronic pain and stroke, surface NMDARs show long-term changes in a brain region-specific and cell-specific manner[19–21]. These findings indicate that NMDAR trafficking is precisely regulated under both physiological and pathological conditions.

Accumulating evidence shows that NMDAR trafficking within the synapse is regulated by post-translational modification of different NMDAR subunits and by complex interactions between NMDARs and a variety of proteins, including PDZ-domain proteins such as postsynaptic density protein 95 (PSD-95) and synapse-associated protein 102[1, 8, 22, 23]. Furthermore, several protein families involved in vesicle trafficking have been shown to participate in the internalization and membrane insertion of NMDARs, such as clathrin and its adaptor AP2 for the internalization of NMDARs[24], and exocyst complex and SNARE proteins (comprising families of membrane-associated proteins, including SNAP25, syntaxin, and synaptobrevin/vesicle-associated membrane protein) for the insertion of NMDARs into the plasma membrane[11, 25–28].

Rab3-interacting molecules (RIMs) are evolutionarily conserved proteins that play critical roles in presynaptic neurotransmitter release[29–31]. RIMs participate in the docking and priming of presynaptic vesicles[32–34], as well as the tethering of vesicles and $Ca^{2+}$ channels[35, 36]. In the present study, we showed that RIM1, a major RIM isoform, was located both pre- and post-synaptically in the mouse hippocampus. RIM1 knockdown in the hippocampal CA1 region not only affected NMDAR-mediated synaptic responses, leaving AMPAR-mediated synaptic responses unaltered, but also impaired LTP and hippocampus-dependent memory. In addition, the RIM1 levels in cultured hippocampal neurons determined both constitutive and regulated NMDAR trafficking, but not constitutive AMPAR trafficking. Furthermore, we found that RIM1 bound to Rab11 via its N-terminus, and knockdown of RIM1 impaired the surface localization of recycling NMDARs. Taken together, our results identify a substantial role for postsynaptic RIM1 in facilitating NMDAR recycling and suggest that this mechanism is important for synaptic function and long-term memory.

## Results

### RIM1 is located both presynaptically and postsynaptically.

Previous work has shown that RIMs form the core of the active zone and mediate the docking and priming of presynaptic vesicles[37]. To test whether RIMs are also involved in postsynaptic vesicle trafficking, we detected the subcellular localization of RIMs by the synaptosome fractionation of mouse cortex[38]. The further digestion of synaptosomes yields an insoluble "PSD-enriched" (synaptic) membrane fraction and a "non-PSD-enriched" (extrasynaptic) membrane fraction[39]. We were able to separate PSD-enriched and non-PSD-enriched membrane as demonstrated by the distribution of the postsynaptic marker PSD-95 in the PSD fraction and that of the presynaptic markers synaptophysin and Rab3 in the non-PSD fraction (Fig. 1a). The AMPA receptor subunit GluA1 was located in both the non-PSD and PSD fractions. Most of the NMDAR subunits GluN2A and GluN2B were located in the PSD fraction, whereas synapsin II, a regulator of neurotransmitter release, and RIM binding protein 2 (RBP2), which couples RIMs to $Ca^{2+}$ channels, were located in the non-PSD fraction (Fig. 1a). Moreover, Rab3 effector RIMs (both RIM1 and RIM2) were located in both the PSD and non-PSD fractions (Fig. 1a). We also performed sucrose gradient centrifugation to obtain the synaptic plasma membrane (SPM) and postsynaptic densities (PSDs) of mouse cortex[40]. As shown in Fig. 1b, PSD-95 was enriched in the PSD fraction, while synaptophysin and Rab3 were not. GluN2A, GluN2B, and GluA1 were located in both the PSD and SPM fractions, whereas synapsin II and RBP2 were located only in the SPM but not the PSD fraction (Fig. 1b). In contrast, RIM1 and RIM2 were located not only in the SPM fraction but also in the PSD fraction, though at lower levels (Fig. 1b). When 2-fold quantities of PSD fraction samples were loaded, RIM1 and PSD-95 were more abundant, while presynaptic synaptophysin, Bassoon, and Munc13 were still not detected (Fig. 1c).

Next, we used pre-embedding immuno-electron microscopy with immunogold staining for RIM1 and observed the distribution of RIM1 in the hippocampal CA1 region. The RIM1 immunoreactivity was mainly located in asymmetric synapses, in both presynaptic and postsynaptic sites. In 410 synapses (from four mice) with nano-gold-labeled RIM1, we found gold particles on 326 (79.5%) of presynaptic (Fig. 1d, left panel; also see Supplementary Fig. 1a–c) and on 189 (46.1%) of postsynaptic sites (Fig. 1d, middle panel; also see Supplementary Fig. 1d–f). In many cases, gold particles were simultaneously located at presynaptic and postsynaptic sites (105, 25.6%) (Fig. 1d, right panel; also see Supplementary Fig. 1g–i). These results indicated that RIM1 is distributed both pre- and post-synaptically in area CA1. To test the specificity of the antibody against RIM1, we injected AAV-hSyn-Cre-GFP into the hippocampus of RIM1[floxed] mice[41]. Three weeks after injection, the hippocampus was sectioned and stained with antibody against RIM1. As shown in Supplementary Fig. 2, GFP-positive cells showed much lower RIM1 fluorescence intensity than GFP-negative cells.

### Postsynaptic RIM1 knockdown impairs NMDAR-mediated transmission.

To identify the postsynaptic role of RIM1 in regulating excitatory synaptic transmission, we generated lentivirus harboring short-hairpin RNAs (shRNAs) with a sequence previously demonstrated to knock down endogenous RIM1 (RIM1 KD)[42]. RIM1 KD specifically inhibited the expression of endogenous RIM1 as determined by quantitative densitometry of immunoblots (control RNAi, 100 ± 8.7%; RIM1 RNAi, 29 ± 9.3%; Supplementary Fig. 3). In contrast, RIM1 KD did not alter the total expression of GluN1, GluN2A, GluN2B, synaptophysin, RIM2, or RBP2 (Supplementary Fig. 3).

To exclude the effect of presynaptic RIM1 on synaptic transmission, we injected lentivirus into the CA1 region of 3-week-old mice to specifically reduce the expression level of RIM1 in that area, while leaving RIM1 in area CA3 unchanged, and then tested the CA3–CA1 Schaffer collateral (SC) synapses

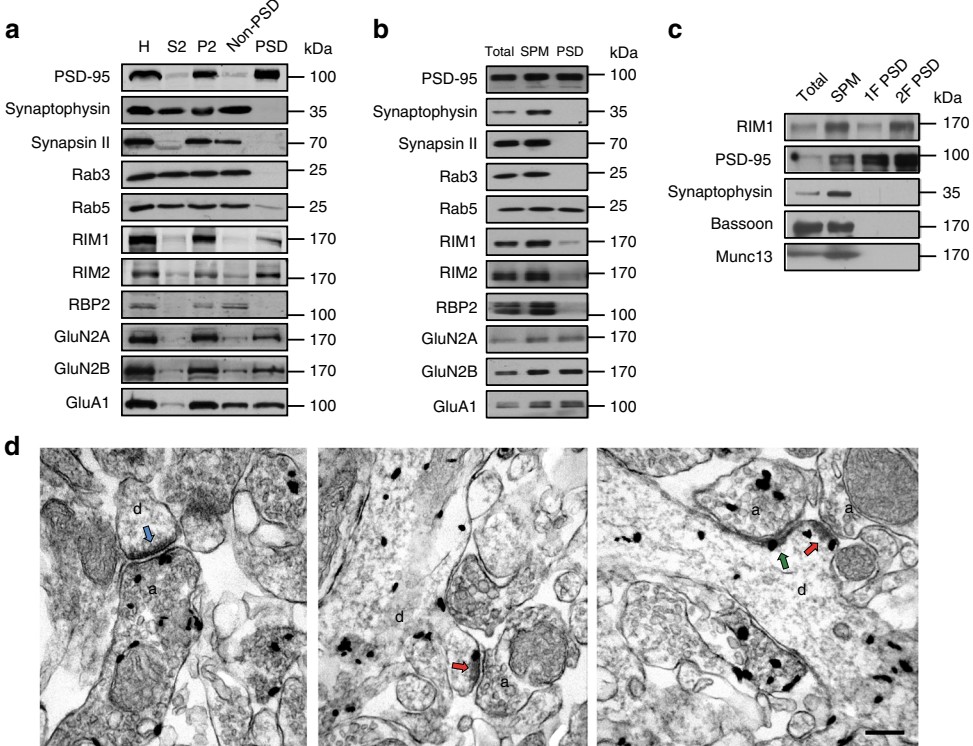

**Fig. 1** RIM1 is localized both presynaptically and postsynaptically. **a** Subcellular fractions of mouse cortex by synaptosome fractionation were probed for the presynaptic marker synaptophysin, the postsynaptic marker PSD-95, the RIM proteins RIM1 and RIM2, the RIM-related protein RBP2, the NMDAR subunits GluN2A and GluN2B, the AMPAR subunit GluA1, and other important synaptic proteins such as synapsin II, Rab3, and Rab5. H homogenates, S2 supernatant after first high-speed centrifugation (12,000×$g$), P2 pellet after the first high-speed centrifugation (12,000×$g$). **b** Subcellular fractions of mouse cortex by sucrose gradient centrifugation were probed for synaptophysin, PSD-95, RIM1, RIM2, RBP2, GluN2A, GluN2B, GluA1, and other important synaptic proteins such as synapsin II, Rab3, and Rab5. Total, homogenates; SPM synaptic plasma membrane fraction; PSD postsynaptic density fraction. **c** Subcellular fractions of mouse cortex by sucrose gradient centrifugation were probed for the presynaptic markers synaptophysin, Munc13, and Bassoon, the postsynaptic marker PSD-95, and RIM1. Total, homogenates; SPM synaptic plasma membrane fraction; PSD postsynaptic density fraction; 1F and 2F are represented as the normal and the 2-fold quantity of the PSD fraction sample, respectively. **d** Electron micrographs of adult mouse hippocampus stained for RIM1 with immunogold (data from 4 mice). Left: a nanogold-labeled axon terminal makes a synapse with an immunonegative dendritic spine. Middle: an immunonegative axon makes a synapse with a nanogold-labeled dendrite. Right: a nanogold-labeled axon and an immunonegative axon make synapses with a nanogold-labeled dendrite; a axon, d dendrite; blue arrow, synapse with nanogold in the presynaptic profile; red arrow, synapse with nanogold in the postsynaptic profile; green arrow, synapse with nanogold in both the pre- and postsynaptic profiles. All arrows point to the PSD. Scale bar, 200 nm

(Fig. 2a). Whole-cell patch-clamp recordings were made from the infected CA1 pyramidal cells in acute slices 10 to 15 days after virus injection (Supplementary Fig. 4a). Paired-pulse facilitation (PPF) is a transient form of plasticity commonly used as a measure of presynaptic function[43] and RIM1 KD had no effect on PPF compared to neurons infected with control RNAi (Fig. 2b). In contrast, when we injected the lentivirus into the CA3 region and recorded PPF from the CA1 pyramidal cells, we found that presynaptic RIM1 KD in area CA3 significantly affected PPF in area CA1 (control RNAi, 2.0 ± 0.1, $n = 9$, 4 mice; RIM1 RNAi, 3.1 ± 0.5, $n = 10$, 4 mice; $p < 0.05$; Supplementary Fig. 4b and c), in accord with the previous finding that presynaptic RIM1 is critical for neurotransmitter release[29–31]. These results are consistent with our hypothesis that RIM1 KD occurs only postsynaptically at the synapses being studied when lentivirus is injected into the CA1 region.

Subsequently, we examined the effect of RIM1 KD on AMPAR-mediated miniature excitatory postsynaptic currents (AMPAR-mEPSCs). Neither the frequency nor the amplitude of mEPSCs was affected by RIM1 KD (mean frequency: control RNAi, 0.8 ± 0.1 Hz, $n = 9$, 5 mice; RIM1 RNAi, 0.8 ± 0.2 Hz, $n = 9$, 5 mice; $p > 0.05$; mean amplitude: control RNAi, 17.7 ± 1.8 pA, $n = 9$, 5 mice; RIM1 RNAi, 17.6 ± 0.7 pA, $n = 9$, 5 mice; $p > 0.05$; Fig. 2c). To explore whether postsynaptic RIM1 KD influences basal NMDAR-mediated synaptic transmission, we calculated the NMDAR/AMPAR ratio. As shown in Fig. 2d, this ratio was significantly lower in the CA1 pyramidal cells with RIM1 KD than in control cells (control RNAi, 0.46 ± 0.02; RIM1 RNAi, 0.21 ± 0.03; $p < 0.001$). RIM1 KD did not alter the current–voltage relationship of NMDAR-mediated EPSCs (Fig. 2e), suggesting little impact on channel properties. We further assessed the role of postsynaptic RIM1 in NMDAR-dependent plasticity, and found that RIM1 KD impaired NMDAR-dependent LTP in the CA1 region (127.7 ± 20.0% of baseline, $n = 8$ cells, 5 mice; Fig. 2f) while control RNAi had no such effect (325.9 ± 45.2%, $n = 8$ cells, 5 mice; Fig. 2f).

Next, we analyzed the NMDAR-mediated EPSCs evoked by various stimulus intensities in the presence of the AMPAR blocker 6-cyano-7-nitroquinoxaline-2, 3-dione (CNQX; 20 µM). The input–output relationship of NMDAR-mediated EPSCs in the CA1 pyramidal cells with RIM1 KD was significantly lower than that in control cells (Fig. 3a). In contrast, RIM1 KD had no effect on the input–output relationship of AMPAR currents (Fig. 3b). We also analyzed the NMDAR-mEPSCs and found that the amplitude was lower in the RIM1 KD group than in the control group (mean amplitude: control RNAi, 8.7 ± 0.4 pA, $n = 7$, 5 mice; RIM1 RNAi, 7.0 ± 0.3 pA, $n = 8$, 6 mice; $p < 0.01$; Supplementary Fig. 4d). The mean inter-event intervals were not influenced (control RNAi, 4.0 ± 0.2 s, $n = 7$, 5 mice; RIM1 RNAi, 4.6 ± 0.3 s, $n = 8$, 6 mice;

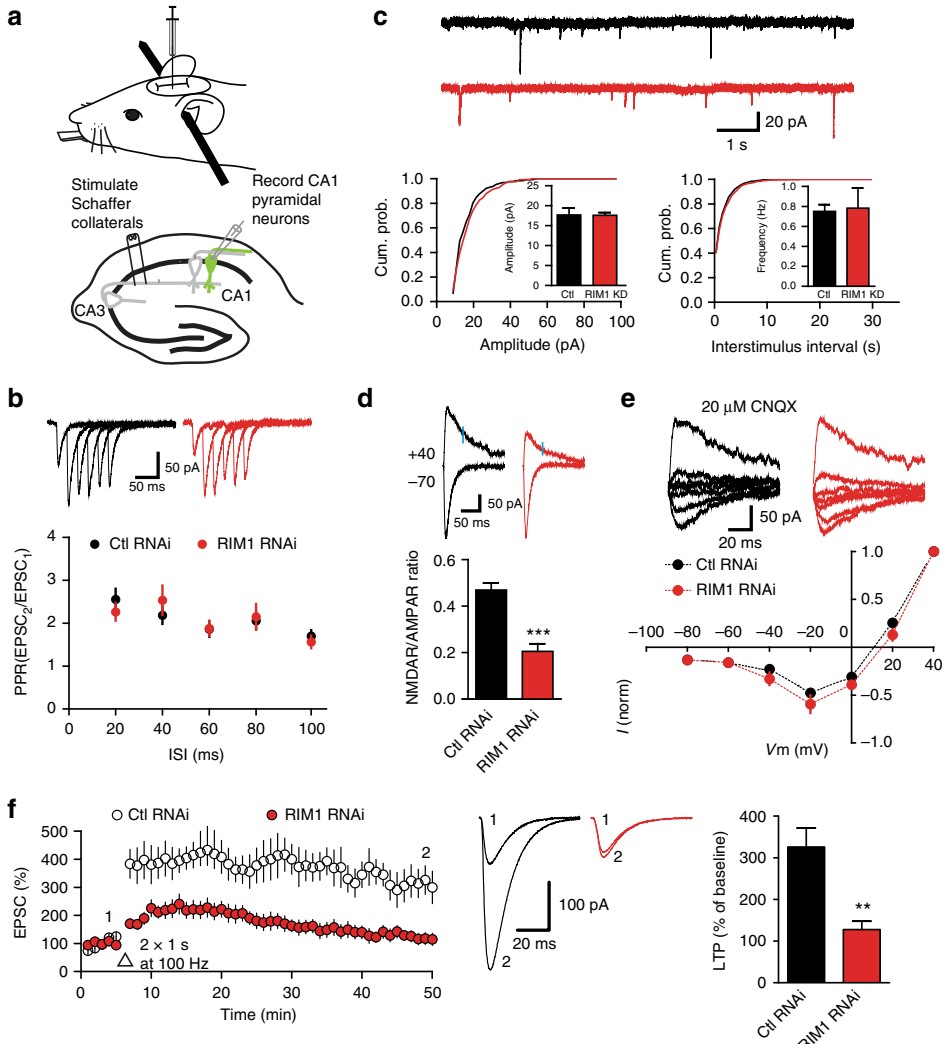

**Fig. 2** Postsynaptic RIM1 KD impairs NMDAR-dependent LTP. **a** Diagram (upper) showing mouse on stereotaxic apparatus for injection of lentivirus. Schematic (lower) illustrating recording from infected neurons in the pyramidal layer of CA1. **b** Paired-pulse facilitation (PPF) analyzed in separate Student's unpaired $t$-tests at different inter-stimulus intervals (ISIs) reveals no difference between KD and control groups; $n = 9$ cells from 4 mice for RIM1 RNAi group and 9 cells from 4 mice for control RNAi group, $p > 0.05$. **c** Mean and cumulative frequency of mEPSCs is unchanged in neurons with RIM1 KD. Short traces of recordings from representative cells are shown above the graph; $n = 9$ cells from 5 mice for RIM1 RNAi group and 9 cells from 5 mice for control RNAi group. The mean amplitude and frequency of mEPSCs are reported as mean ± s.e.m.; $p > 0.05$, unpaired two-tailed $t$-test (referred to as "$t$-test" throughout except as noted). The cumulative frequency of amplitude and inter-event intervals were analyzed with the Kolmogorov–Smirnov test, $p > 0.05$. **d** Ratio of NMDAR- to AMPAR-mediated EPSCs is significantly decreased by RIM1 KD. Representative EPSCs recorded at −70 mV and +40 mV are shown; $n = 11$ cells from 7 mice for RIM1 RNAi group and 13 cells from 7 mice for control RNAi group. Data are reported as mean ± s.e.m. $t$-test; ***$p <$ 0.001. **e** RIM1 KD has no effects on the current–voltage relationship of NMDAR-mediated EPSCs. Short traces of recordings from representative cells are shown above the graph; $n = 7$ cells from 4 mice for RIM1 RNAi group and 9 cells from 4 mice for control RNAi group. The data are reported as mean ± s.e.m. and analyzed in separate $t$-tests at each voltage, $p > 0.05$. **f** Postsynaptic RIM1 KD impairs NMDAR-dependent LTP. Left LTP time course; middle, representative traces; right, summary graphs of LTP magnitude; $n = 8$ cells from 5 mice for RIM1 RNAi group and 8 cells from 5 mice for control RNAi group. The data are reported as mean ± s.e.m. $t$-test; **$p < 0.01$

$p > 0.05$; Supplementary Fig. 4d), while the cumulative distributions showed slight rightward shifts ($p = 0.0426$).

Next, we conducted simultaneous dual whole-cell recordings from infected and neighboring uninfected pyramidal cells in the CA1 region (Fig. 3c). Postsynaptic RIM1 KD resulted in a lower amplitude of NMDAR-mediated EPSCs at SC-CA1 synapses in the infected neurons than in neighboring uninfected neurons, but no change in their decay time (control RNAi, $182.6 \pm 19.3$ ms, $n = 10$ cells, 5 mice; RIM1 RNAi, $177.6 \pm 16.2$ ms, $n = 10$ cells, 5 mice; $p > 0.05$; Fig. 3d). The amplitude of AMPAR-mediated EPSCs did not differ between RIM1 KD neurons and neighboring GFP-negative neurons (Fig. 3e). Control RNAi had no effect on

the amplitude or decay time of NMDAR-mediated EPSCs (decay time: control RNAi, $173.5 \pm 10.9$ ms, $n = 9$ cells, 4 mice; RIM1 RNAi, $167.0 \pm 13.1$ ms, $n = 9$ cells, 4 mice; $p > 0.05$; Fig. 3f) or the amplitude of AMPAR-mediated EPSCs (Fig. 3g) at SC-CA1 synapses. Taken together, these data indicate that postsynaptic RIM1 KD leads to the downregulation of NMDAR-mediated transmission, while leaving that of AMPARs unaltered.

**RIM1 KD in area CA1 impairs hippocampus-dependent memory.** We then asked whether RIM1 in the CA1 region is involved in hippocampus-dependent learning and memory.

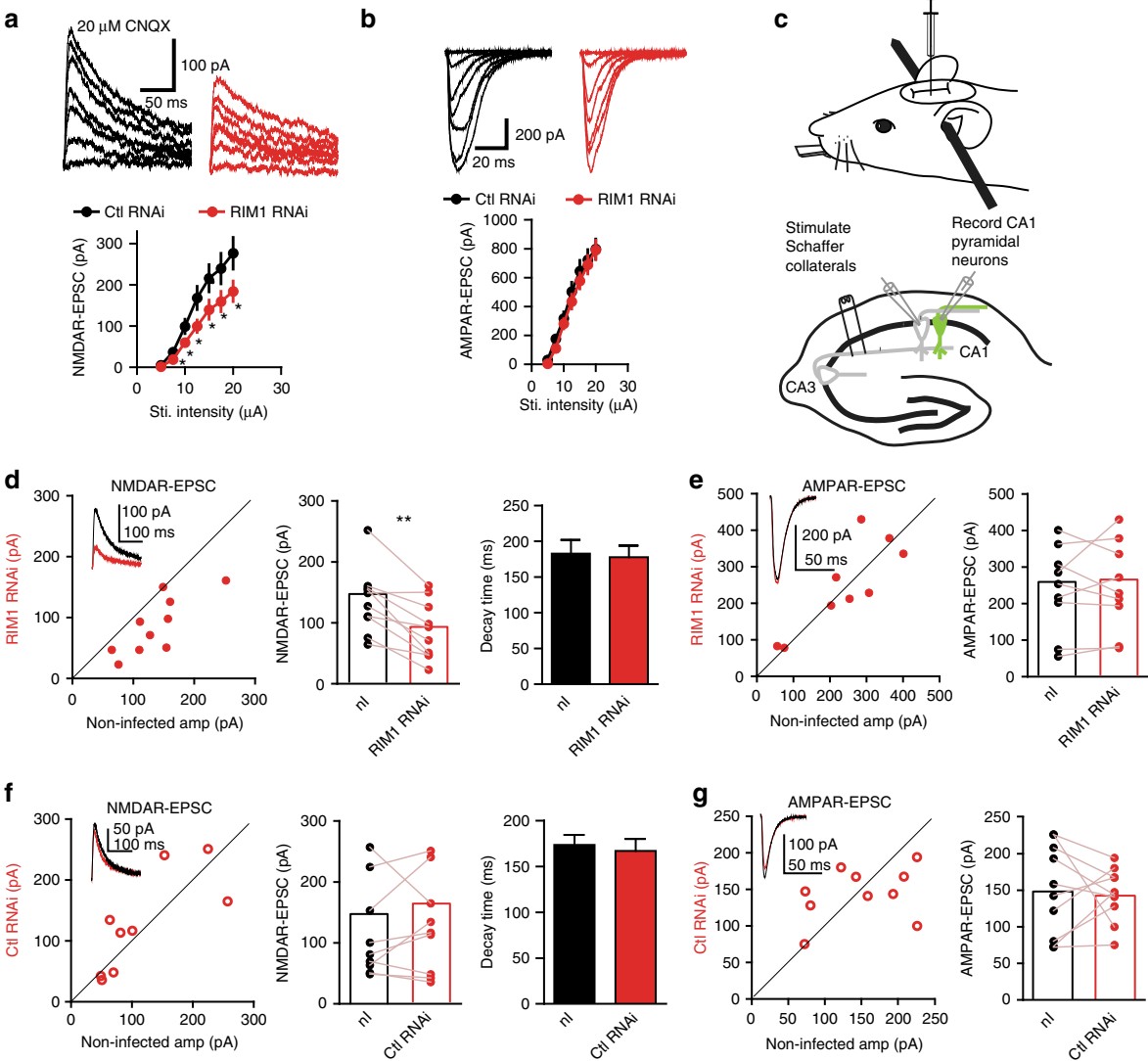

**Fig. 3** Postsynaptic RIM1 KD impairs NMDAR-mediated transmission. **a** NMDAR input/output curve in CA1 pyramidal neurons reveals a significant effect of RIM1 RNAi on the amplitude of NMDAR current across stimulation intensities. EPSCs from two representative cells are shown above the graph; $n = 9$ cells from 4 mice for RIM1 RNAi group and 8 cells from 4 mice for control RNAi group. The data were analyzed in separate $t$-tests at each stimulus intensity; *$p < 0.05$. **b** Measurement of AMPAR-EPSCs elicited by isolated stimuli applied with increasing strength to obtain input/output relationships in neurons with control RNAi or RIM1 RNAi. $n = 10$ cells from 5 mice for each group. The data were analyzed in separate $t$-tests at each stimulus intensity; $p > 0.05$. **c** Diagram (upper) showing mouse on stereotaxic apparatus for injection of lentivirus. Schematic (lower) illustrating paired recordings from infected and neighboring uninfected (control) neurons in the pyramidal layer of CA1. **d** Postsynaptic RIM1 KD reduces the amplitude of NMDAR-mediated EPSCs at SC-CA1 synapses ($n = 10$ pairs of neurons from 5 mice. Paired $t$-test; **$p < 0.01$) but does not modify their decay time compared to neighboring uninfected neurons ($t$-test, $p > 0.05$). Sample traces are shown in the inset (red lines, infected neurons; black lines, neighboring non-infected neurons); nl non-infected. **e** Postsynaptic RIM1 KD does not change the amplitude of AMPAR-mediated EPSCs at SC-CA1 synapses compared to neighboring uninfected neurons; $n = 9$ pairs of neurons from 5 mice. Paired $t$-test, $p > 0.05$. **f** Control RNAi does not affect NMDAR-mediated EPSCs at SC-CA1 synapses (amplitude and decay time), as compared to neighboring uninfected neurons. Sample traces are shown in the inset; $n = 9$ pairs of neurons from 4 mice. Paired $t$-test, $p > 0.05$. **g** Control RNAi does not modify the amplitude of AMPAR-mediated EPSCs at SC-CA1 synapses; $n = 10$ pairs of neurons from 6 mice, $p > 0.05$, paired $t$-test. In all panels, bar graphs and individual points represent the mean ± s.e.m.; *$p < 0.05$, **$p < 0.01$, ***$p < 0.001$

Lentivirus with RIM1 KD or a control sequence was injected into the CA1 region of 8-week-old mice, and their performance in cognitive tasks was assessed 2 weeks later. First, mice with RIM1 KD in the CA1 region were tested in an object location task that is specifically dependent on the hippocampus (Fig. 4a)[44]. In the acquisition phase, all mice showed comparable performance and spent similar amounts of time exploring the two objects (control RNAi, $50.7 ± 1.4\%$, $n = 10$ mice; RIM1 RNAi, $49.9 ± 3.3\%$, $n = 10$ mice; $p > 0.05$; Fig. 4b). In the retrieval phase, control mice spent more time investigating the object in the unfamiliar location, whereas RIM1 KD mice failed to show a preference for the

relocated object (control RNAi, $64.7 ± 3.7\%$, $n = 10$ mice; RIM1 RNAi, $49.0 ± 2.5\%$, $n = 10$ mice; $p < 0.05$; Fig. 4c). Given that RIM1 was specifically knocked down in the CA1 region, we used a temporal order memory task (Fig. 4d) that is sensitive to area CA1 but not CA3[45, 46]. In the retrieval phase, control mice preferred the object explored earlier to that explored last, whereas the performance of RIM1 KD mice was impaired in this task (control RNAi, $68.4 ± 2.9\%$, $n = 10$ mice; RIM1 RNAi, $51.3 ± 3.6\%$, $n = 10$ mice; $p < 0.05$; Fig. 4e). We also tested the mice in a novel object preference task (Fig. 4f) that is dependent on the perirhinal cortex but not the hippocampus[44]. Both control and RIM1 KD mice

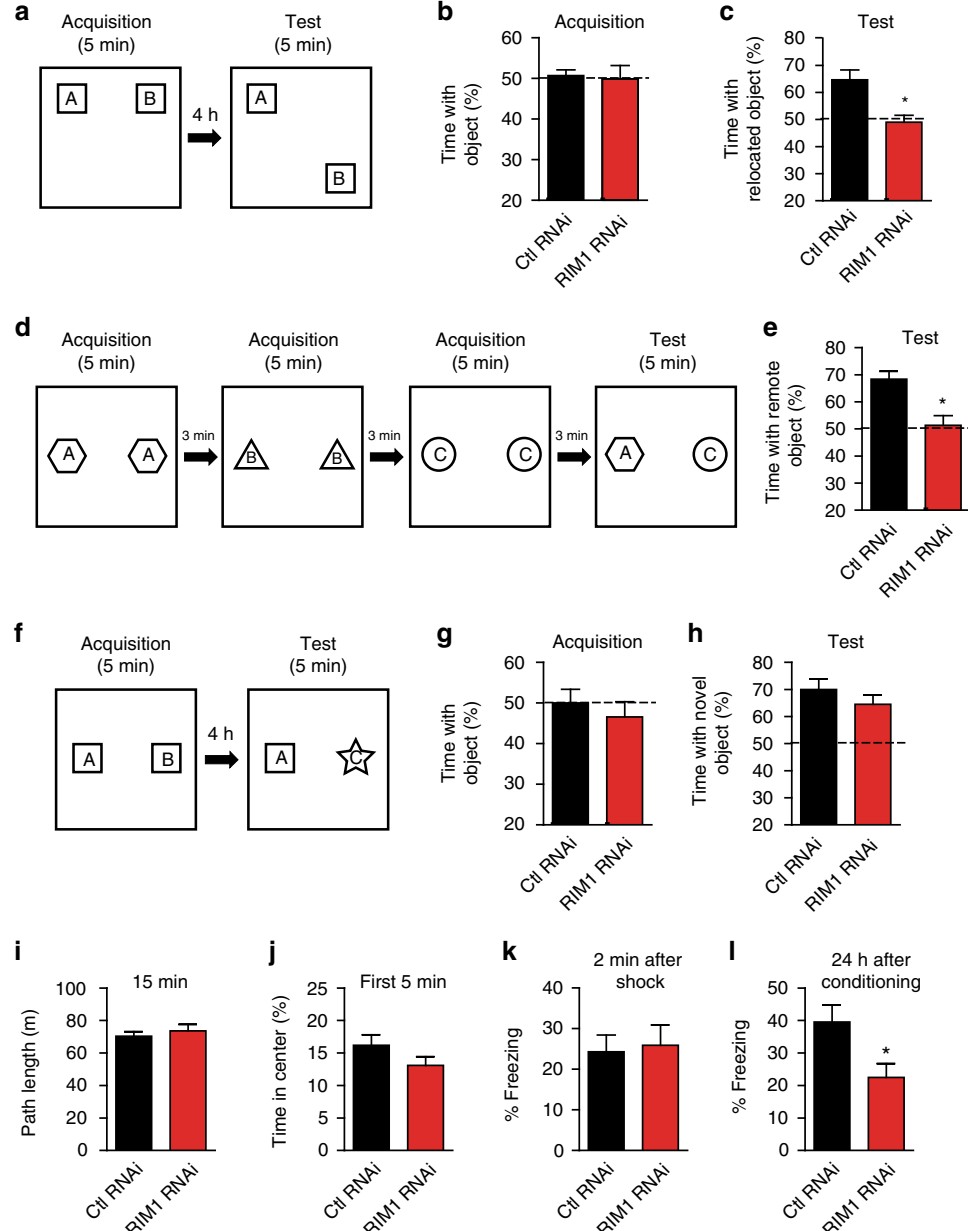

**Fig. 4** RIM1 KD in the CA1 region impairs hippocampus-dependent memory. **a** Experimental designs for the object location task. **b** In the acquisition phase of the object location task, RIM1 KD and control mice spend similar time exploring the two objects; $n = 10$ mice/group, t-test, $p > 0.05$. **c** In the test phase of the object location task, RIM1 KD mice fail to show preference for the relocated object; $n = 10$ mice/group, t-test, *$p < 0.05$. **d** Experimental design for the temporal order memory task. **e** In the temporal order memory task, control mice prefer the object explored early to that explored last, whereas mice with RIM1 KD have impaired performance in this task; $n = 10$ mice/group, t-test, *$p < 0.05$. **f** Experimental design for the novel object preference task. In both the acquisition phase (**g**) and the test phase (**h**) of the novel object preference task, RIM1 KD and control mice spend similar time exploring the two objects; $n = 10$ mice/group, t-test, $p = 0.34$. **i** In the open field test, RIM1 KD and control mice show similar locomotor activity during 15 min of open field exploration; $n = 10$ mice/group, t-test, $p > 0.05$. **j** RIM1 KD mice and control mice spend similar time in the center of the open area during the first 5 min of exploration; $n = 10$ mice/group, t-test, $p > 0.05$. **k** In contextual fear conditioning, there is no significant difference between RIM1 KD and control mice in freezing responses during the last 2 min of training; $n = 10$ mice/group, t-test, $p > 0.05$. **l** Twenty-four hours after conditioning, RIM1 KD mice show a significant reduction in the time spent freezing; $n = 10$ mice/group. Bar graphs and individual points represent the mean ± s.e.m.; t-test, *$p < 0.05$

explored the novel object more frequently than the known one in the retrieval phase, indicating normal perirhinal cortex-dependent memory (control RNAi, 78.4 ± 5.0%, $n = 10$ mice; RIM1 RNAi, 71.3 ± 5.1%, $n = 10$ mice; $p = 0.34$; Fig. 4g, h). In addition, RIM1 KD did not evoke noticeable changes in locomotion (control RNAi, 23.4 ± 0.9 m, $n = 10$ mice; RIM1 RNAi, 24.5 ± 1.3 m, $n = 10$ mice; $p > 0.05$; Fig. 4i), and had only weak effects that did not reach a 0.05 level of significance on the time

spent in the center of the open arena during the first 5 min of exploration (control RNAi, 16.2 ± 1.6%, $n = 10$ mice; RIM1 RNAi, 13.1 ± 1.3%, $n = 10$ mice; $p = 0.15$; Fig. 4j). Together, these results suggest that RIM1 in the CA1 region is required for hippocampal CA1-dependent recognition memory.

To further clarify the role of RIM1 in emotional learning and memory, we assessed hippocampus-dependent contextual fear conditioning in RIM1 KD and control mice. There was no

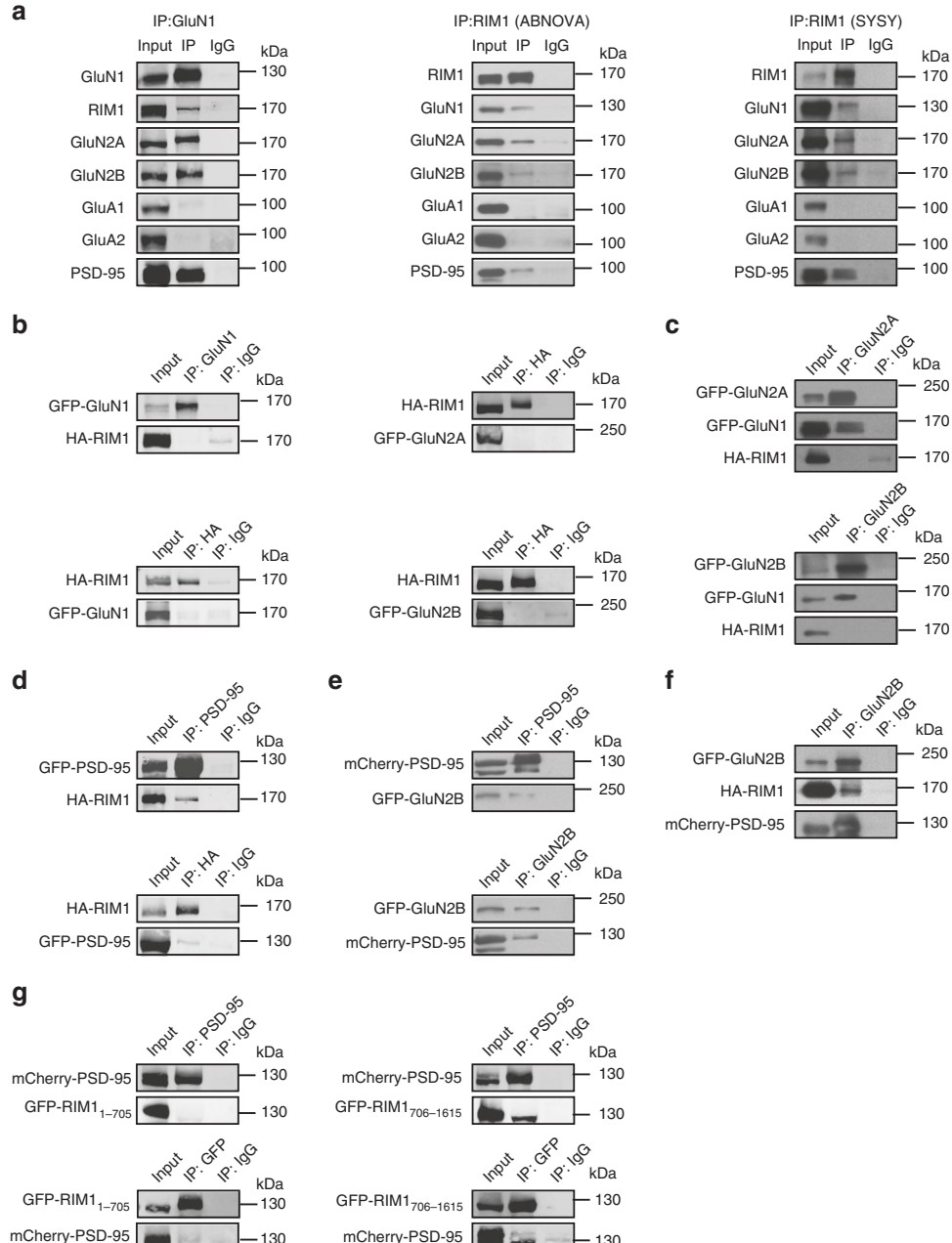

**Fig. 5** RIM1 is a binding partner of the NMDAR complex in mouse brain. **a** RIM1 is within the NMDAR complex, but not the AMPAR complex. Extracts from adult mouse cortex were immunoprecipitated with antibodies against GluN1 (left) or RIM1 (middle: antibody from ABNOVA; right, antibody from SYSY) and blotted with corresponding antibodies. IP immunoprecipitation. **b** RIM1 has no interaction with NMDAR subunits in transfected HEK293T cells. Extracts of HEK293T cells transfected with HA-RIM1 and GFP-GluN1 (left), GFP-GluN2A (upper right), or GFP-GluN2B (lower right) were immunoprecipitated with GluN1 or HA antibody and blotted with the corresponding antibodies. **c** Extracts of HEK293T cells co-transfected with HA-RIM1, GFP-GluN1, and GFP-GluN2A (upper), or HA-RIM1, GFP-GluN1, and GFP-GluN2B (lower) were immunoprecipitated with GluN2A or GluN2B antibody and blotted with the corresponding antibodies. **d** RIM1 interacts with PSD-95 in transfected HEK293T cells. **e** GluN2B interacts with PSD-95 in transfected HEK293T cells. **f** Extracts of HEK293T cells co-transfected with HA-RIM1, GFP-GluN2B, and mCherry-PSD-95 were immunoprecipitated with GluN2B antibody and blotted with the corresponding antibodies. **g** Different RIM1 fragments tagged with GFP (GFP-RIM1$_{1-705}$ or GFP-RIM1$_{706-1615}$) were cotransfected with mCherry-PSD-95 into HEK293 cells and the interaction between these two proteins was tested by immunoprecipitation

significant difference between groups in freezing responses immediately after training (control RNAi, 24.3 ± 4.1%, $n = 10$ mice; RIM1 RNAi, 25.9 ± 5.0%, $n = 10$ mice; $p > 0.05$; Fig. 4k), suggesting that RIM1 KD does not impair the shock-induced freezing response. However, RIM1 KD mice showed a significant reduction in the time spent freezing at 24 h after conditioning (control RNAi, 39.5 ± 5.3%, $n = 10$ mice; RIM1 RNAi, 22.5 ± 4.3%, $n = 10$ mice; $p < 0.05$; Fig. 4l), indicating an impairment in contextual fear memory.

**RIM1 is a binding partner of the NMDAR complex**. We used biochemical assay to detect whether endogenous RIM1 is associated with NMDARs or AMPARs in cortical lysates from mice. We performed immunoprecipitation under non-denaturing conditions of receptor solubilization with two antibodies raised against different epitopes of RIM1 and found that both of them coimmunoprecipitated PSD-95 and the NMDAR subunits GluN1, GluN2A, and GluN2B, but not the AMPAR subunits

GluA1 or GluA2 (Fig. 5a, middle and right panels). In addition, antibodies against GluN1 coimmunoprecipitated RIM1, GluN2A, GluN2B, and PSD-95, but not GluA1 (Fig. 5a, left panel). To exclude the possibility that non-solubilized membrane fragments were present before immunoprecipitation, we cleared the tissue extracts by ultracentrifugation ($100,000 \times g$ for 1 h). As shown in Supplementary Fig. 5, both PSD-95 and GluN2B coimmuno-precipitated RIM1, and RIM1 coimmunoprecipitated GluN2B. These results indicate that RIM1 is a specific binding partner of the NMDAR multi-protein complex in vivo, but not that of the AMPAR.

We further examined the developmental profile of RIM1 and NMDARs in the rat cortex and hippocampus and found that the expression of RIM1 was low on postnatal day 1 (P1) and gradually increased through P20 in both hippocampus and cortex (see Supplementary Fig. 6).

To investigate the interaction of RIM1 with different NMDAR subunits, we cotransfected HA-tagged RIM1 (HA-RIM1) with GFP-tagged NMDAR subunits (GFP-GluN1, GFP-GluN2A, or GFP-GluN2B) into HEK293T cells. The HA antibody precipi-tated HA-RIM1 efficiently but was unable to coimmunoprecipi-tate GFP-GluN1, GFP-GluN2A, or GFP-GluN2B (Fig. 5b). Similarly, GluN1 antibody precipitated GFP-GluN1 effectively but coimmunoprecipitated no HA-RIM1 (Fig. 5b). NMDAR subunits expressed alone in HEK293 cells are retained in the ER. Next, GluN1/GluN2A/HA-RIM1 or GluN1/GluN2B/HA-RIM1 were co-transfected into HEK293 cells. However, as shown in Fig. 5c, no interaction was detected between HA-RIM1 and NMDAR subunits. Taken together, these results indicate that RIM1 interacts indirectly with NMDARs.

PSD-95 is a key scaffolding protein associated with NMDARs, and we also observed that both PSD-95 and RIM1 were within the NMDAR complex (Fig. 5a). We then co-transfected GFP-PSD-95 and HA-RIM1 into HEK293T cells and found an interaction between PSD-95 and RIM1 (Fig. 5d). In addition, an interaction between PSD-95 and GluN2B was detected when mCherry-PSD-95 and GFP-GluN2B were co-transfected into HEK293 cells (Fig. 5e). To test whether PSD-95 acts as a bridge coupling NMDARs with RIM1, we cotransfected HA-RIM1, GFP-GluN2B, and mCherry-PSD-95 into HEK293 cells and found that GluN2B coimmunoprecipitated not only PSD-95, but also RIM1 (Fig. 5f). We further examined the interaction with RIM1 fragments composed of either its N-terminal Rab3-binding sequence and PDZ domain ($RIM1_{1-705}$), or of its C-terminal fragments containing the $C_2A$ and $C_2B$ domains and the RIM-binding protein sequence ($RIM1_{706-1615}$). As shown in Fig. 5g, the C-terminal $RIM1_{706-1615}$ interacted with PSD-95, whereas the N-terminal $RIM1_{1-705}$ did not. Taken together, these results suggest that RIM1 is recruited to the NMDAR complex by PSD-95.

**RIM1 levels affect the surface localization of NMDARs**. To further investigate the effect of RIM1 on surface NMDAR levels, we knocked down endogenous RIM1 via transfection with plas-mids containing the RIM1 shRNA sequence in cultured hippo-campal neurons at DIV6. The $Ca^{2+}$-phosphate method with a transfection efficiency of ~5%[47, 48] was used to minimize inter-ference by presynaptic effects. After transduction, the effect of RIM1 KD on the surface localization of native NMDARs was assessed using immunostaining with an antibody against the extracellular N-terminus of GluN2B. We focused on the NMDAR subtype containing GluN2B (GluN2B/NMDAR), since GluN2B/NMDARs tend to recycle after internalization[49] and GluN2B plays a dominant role in the trafficking of heterotrimeric NMDARs containing both GluN2A and GluN2B subunits (GluN2A/GluN2B/NMDARs)[50]. To confirm the specificity of the

antibody against the extracellular N-terminus of GluN2B, we transfected GluN2B into HEK293T cells and detected a clear band at 170 kDa (Supplementary Fig. 7a). We also tested the colocalization of surface-labeled GluN2B with PSD-95 and observed that the surface GluN2B was partially colocalized with PSD-95 ($31 \pm 1\%$, Supplementary Fig. 7b). Moreover, GluN2B knockdown significantly reduced the surface abundance of GluN2B in cultured hippocampal neurons (Supplementary Fig. 7c). As shown in Fig. 6a, RIM1 KD led to a lower intensity of surface-stained native GluN2B than control RNAi (control RNAi, $100 \pm 4\%$, $n = 45$ neurons; RIM1 RNAi: $81 \pm 3\%$, $n = 29$ neurons; $p < 0.05$). In addition, when RIM1 KD rescue plasmids were transfected into hippocampal neurons, the intensity of surface-stained GluN2B did not differ from that of the control RNAi (RIM1 KD rescue, $108 \pm 10\%$, $n = 26$ neurons; Fig. 6a). The distribution of synaptophysin and PSD-95 was first observed 4 days after transfection, and RIM1 KD had no effect on the cluster density of these two proteins compared to control RNAi plasmids, which indicates that no massive loss of synapses occurs under such conditions (synaptophysin: control RNAi, $100 \pm 5\%$, $n = 31$ neurons; RIM1 RNAi, $92 \pm 4\%$, $n = 27$ neurons; $p > 0.05$; PSD-95: control RNAi, $100 \pm 5\%$, $n = 44$ neurons; RIM1 RNAi, $100 \pm 10\%$, $n = 30$ neurons; $p > 0.05$; Supplementary Fig. 8a and 8b).

We also investigated the effects of RIM1 KD on the surface localization of AMPARs using a specific antibody against the N-terminus of the GluA1 subunit, finding that RIM1 KD had no effect on the intensity of surface-stained native GluA1 (control RNAi, $100 \pm 5\%$, $n = 43$ neurons; RIM1 RNAi, $98 \pm 3\%$, $n = 83$ neurons; $p > 0.05$; Supplementary Fig. 8c).

We recorded NMDA/glycine-evoked NMDAR currents in hippocampal neurons transfected with RIM1 KD plasmids, control RNAi plasmids, and RIM1 KD rescue plasmids. RIM1 KD markedly reduced the currents ($504.3 \pm 53.4$ pA, $n = 11$ neurons; $p < 0.05$; Fig. 6b), while control RNAi had no such effect and the RIM1 KD rescue plasmid completely reversed the RIM1 KD-induced decrease of NMDAR currents (control RNAi, $826.2 \pm 94.6$ pA, $n = 11$ neurons; RIM1 KD rescue, $912.3 \pm 58.4$ pA, $n = 26$ neurons; Fig. 6b). Meanwhile, non-NMDA currents evoked in hippocampal neurons transfected with RIM1 KD plasmids did not differ from those in neurons transfected with control RNAi plasmids (control RNAi, $911 \pm 159.5$ pA, $n = 13$ neurons; RIM1 RNAi, $815 \pm 65.6$ pA, $n = 20$ neurons; $p > 0.05$; Supplementary Fig. 8d).

Finally, we infected cultured hippocampal neurons with virus harboring RIM1 RNAi or control RNAi and evaluated the surface localization of NMDARs using surface biotinylation assays. RIM1 KD led to a significant decrease in the levels of surface biotinylated GluN1 compared to control RNAi (see Supplemen-tary Fig. 8e, RIM1 RNAi, $86 \pm 1.8\%$ of control RNAi; $n = 3$; $p < 0.05$). Taken together, these data indicate that downregulation of RIM1 reduces the surface localization and function of native NMDARs in hippocampal neurons.

We then examined how overexpression of GFP-tagged RIM1 (GFP-RIM1) plasmids in cultured hippocampal neurons affects the surface localization of GluN2B/NMDARs. The intensity of surface GluN2B expression was significantly higher in the presence of GFP-RIM1 than in the GFP control (GFP vector, $100 \pm 4.7\%$, $n = 35$ neurons; GFP-RIM1, $123 \pm 8.8\%$, $n = 20$ neurons; $p < 0.05$; Fig. 6c), indicating that the RIM1 level influences the surface localization of NMDARs in cultured hippocampal neurons.

**RIM1 participates in activity-regulated NMDAR trafficking**. We further determined whether RIM1 participates in activity-

regulated NMDAR trafficking. Here, cultured hippocampal neurons were treated with forskolin and rolipram (FSK/Rol), which have been shown to increase the surface localization of NMDARs[19]. When control RNAi plasmids were transfected, treatment with FSK/Rol significantly increased the surface expression of GluN2B/NMDARs (dimethylsulfoxide (DMSO), $100 \pm 8.8\%$, $n = 27$ neurons; FSK/Rol, $131 \pm 5.7\%$; $n = 51$ neurons; $p < 0.01$; Fig. 6d). In contrast, RIM1 KD impaired the upregulation of surface NMDARs induced by FSK/Rol (DMSO, $100 \pm 7\%$, $n = 15$ neurons; FSK/Rol, $102 \pm 6\%$, $n = 25$ neurons; $p > 0.05$; Fig. 6e), indicating that RIM1 is also involved in activity-regulated NMDAR trafficking.

**RIM1 modulates NMDAR recycling in hippocampal neurons.** The above results showed that RIM1 KD decreased the surface localization of NMDARs, which could be the result of either enhanced internalization of NMDARs from the plasma membrane or impaired insertion of NMDARs into the plasma membrane. We performed endocytosis and recycling assays of GluN2B/NMDARs using an antibody against the N-terminus of GluN2B. Cultured hippocampal neurons were transfected with RIM1 RNAi or control RNAi plasmids at DIV5-7 and observed at DIV9-11. No significant change in GluN2B/NMDAR endocytosis was detected when RIM1 was downregulated (control RNAi, $1 \pm 0.06$, $n = 45$ neurons; RIM1 RNAi, $1.1 \pm 0.1$, $n = 30$ neurons; $p > 0.05$; Fig. 7a). However, RIM1 KD inhibited re-expression of the internalized GluN2B/NMDARs back to the plasma membrane (control RNAi, $1 \pm 0.05$, $n = 51$ neurons; RIM1 RNAi, $0.8 \pm 0.05$; $n = 24$ neurons; $p < 0.05$; Fig. 7b). These data suggest that the decrease of surface-localized NMDARs after RIM1 KD is probably due to impaired receptor recycling.

**RIM1 binds to Rab11 through its N-terminus.** Presynaptic RIM1 operates as an Rab3 effector[31]. To test the possibility that RIM1 functions in a similar way at the postsynaptic site, we explored the relationship of RIM1 with the Rab family of small GTPases involved in the recycling of postsynaptic receptors. Rab11 and Rab4 participate in NMDAR recycling[51, 52], but previous work has shown that RIM1 has no direct interaction with Rab4 in rat brain homogenate[31]. Therefore, we focused on Rab11 in the subsequent experiments. In cortical homogenates from mice, Rab11 coimmunoprecipitated with NMDAR subunits

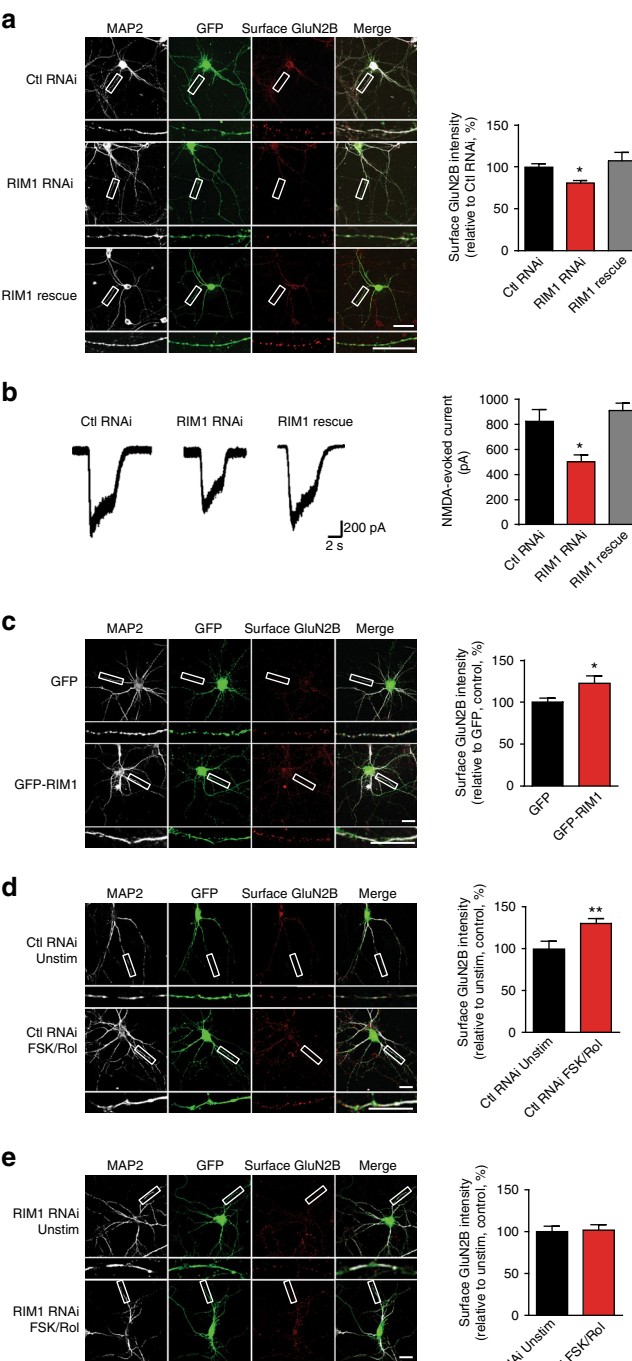

**Fig. 6** RIM1 is involved in both constitutive and activity-dependent NMDAR trafficking. **a** RIM1 KD significantly decreases the surface level of GluN2B/NMDARs. Left: cultured hippocampal neurons were transfected with GFP-tagged control RNAi plasmid (control RNAi, $n = 45$ neurons from three independent cultures), GFP-tagged RIM1 shRNA plasmid (RIM1 RNAi, $n = 29$ neurons from three independent cultures), or GFP-tagged RIM1 shRNA-resistant plasmid (RIM1 rescue, $n = 26$ neurons from three independent cultures), and surface GluN2B was detected by live cell-surface staining. Scale bar, 20 μm. Right: statistical analysis of surface GluN2B intensity. One-way ANOVA with Dunnett's post test, *$p < 0.05$. **b** RIM1 KD significantly reduces the evoked NMDAR currents in cultured cortical neurons. Left: representative recordings of NMDAR currents evoked by NMDA together with glycine. Left: representative traces of NMDA-evoked currents from neurons transfected with control RNAi ($n = 11$ neurons from three independent cultures), RIM1 RNAi ($n = 11$ neurons from three independent cultures), or RIM1 KD rescue ($n = 26$ neurons from three independent cultures) plasmids. Right: statistical analysis of NMDAR-evoked currents. One-way ANOVA with Dunnett's post test, *$p < 0.05$. **c** RIM1 overexpression significantly increases the surface GluN2B/NMDARs levels. Left: representative images of surface GluN2B staining in cultured hippocampal neurons transfected with GFP-RIM1 ($n = 20$ neurons from three independent cultures) or pEGFP-N1 ($n = 35$ neurons from three independent cultures). Scale bar, 20 μm. Right: statistical analysis of surface GluN2B intensity; $t$-test, *$p < 0.05$. **d** Treatment with Forskolin (20 μM) and Rolipram (0.1 μM) for 30 min (FSK/Rol, $n = 51$ neurons from three independent cultures) significantly increases the surface localization of endogenous GluN2B in cultured hippocampal neurons transfected with control RNAi plasmid compared with DMSO ($n = 27$ neurons from three independent cultures). Scale bar, 20 μm; $t$-test, **$p < 0.01$. **e** Treatment with FSK/Rol ($n = 25$ neurons from three independent cultures) does not alter the surface localization of endogenous GluN2B in neurons transfected with RIM1 RNAi plasmid compared with DMSO ($n = 15$ neurons from three independent cultures). Scale bar, 20 μm; $t$-test, $p > 0.05$

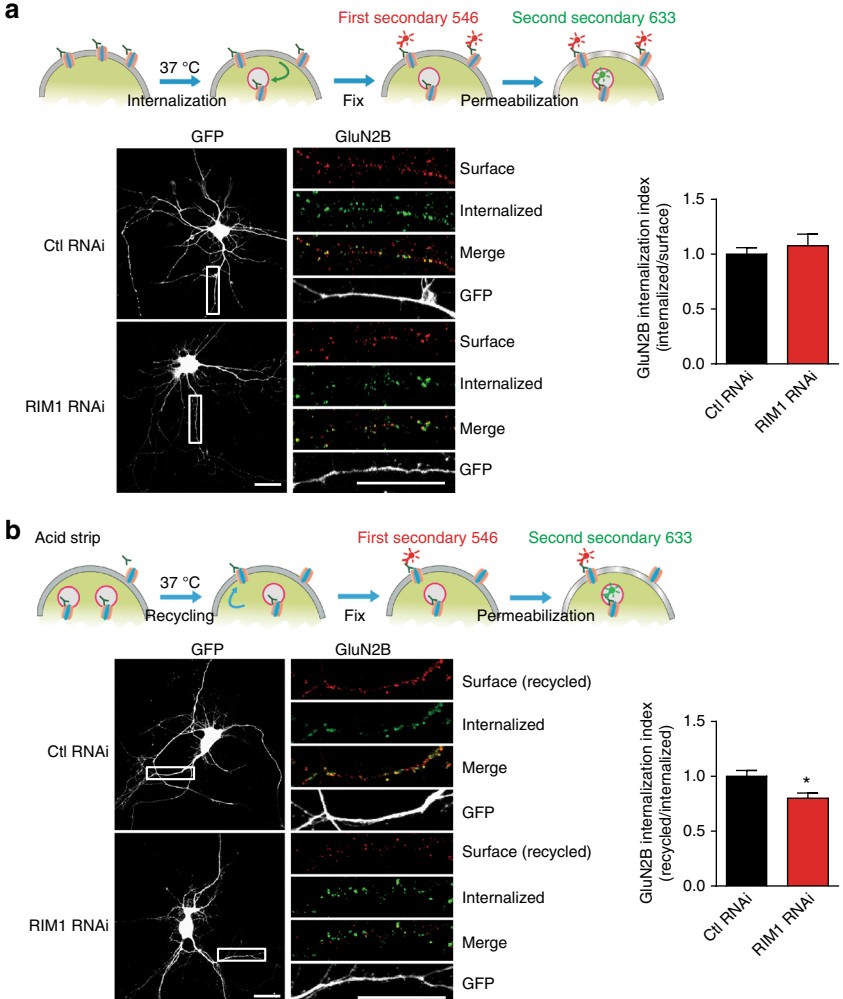

**Fig. 7** RIM1 KD affects the recycling of GluN2B/NMDARs, but not their internalization. **a** RIM1 KD has no effect on the endocytosis of GluN2B/NMDARs. Upper panel: timeline of the endocytosis experiments. Left: internalization of GluN2B/NMDARs in hippocampal neurons transfected with RIM1 RNAi plasmids ($n = 30$ neurons from three independent cultures) or control RNAi plasmids ($n = 45$ neurons from three independent cultures). Red, surface GluN2B/NMDARs; green, internalized GluN2B/NMDARs. Scale bar, 20 μm. Right: statistical analysis of internalized/surface ratio of GluN2B/NMDARs; *t*-test, $p > 0.05$. **b** RIM1 KD significantly decreases the recycling of GluN2B/NMDARs. Upper: timeline of the recycling experiments. Left: recycling of GluN2B/NMDARs in hippocampal neurons transfected with RIM1 RNAi plasmids ($n = 24$ neurons from three independent cultures) or control RNAi plasmids ($n = 51$ neurons from three independent cultures). Red, recycling GluN2B/NMDARs; green, internalized GluN2B/NMDARs. Scale bar, 20 μm. Right: statistical analysis of recycled/internalized ratio of GluN2B/NMDARs; *t*-test, *$p < 0.05$

GluN1 and GluN2B and PSD-95, suggesting that these proteins are partners in the same protein complex (Fig. 8a–c).

We then tested whether the N-terminus of RIM1 containing the alpha helix and the $Zn^{2+}$-finger domain determines the interaction between RIM1 and Rab11. Here, glutathione S-transferase (GST) fusion proteins of the N-terminus of RIM (GST-RIM1$_{1-399}$) were immobilized on glutathione beads and incubated with solubilized brain proteins in the presence of GTP-γS. Consistent with previous findings, Rab3 bound to immobilized GST-RIM1$_{1-399}$, whereas Rab5 did not (Fig. 8d). Rab11 also bound to GST-RIM1$_{1-399}$ (Fig. 8d), indicating that RIM1 interacts directly with Rab11 via its N-terminus. We further focused on the N-terminal alpha helix of RIM1 (RIM1$_{1-55}$) which has been reported to determine the interaction of RIM1 and Rab3[53, 54]. However, GST-RIM1$_{1-55}$ was coimmunoprecipitated with Rab3, but not with Rab5 or Rab11 (Fig. 8e), indicating that the domain mediating the Rab11–RIM1 interaction is different from that mediating the Rab3–RIM1 interaction.

We used three-dimensional structured illumination microscopy (3D-SIM) to examine the colocalization of NMDARs with Rab11. RIM1 KD had no effect on the cluster density of GluN2B (control RNAi, $2.9 \pm 0.3$, $n = 37$ neurons; RIM1 RNAi, $2.9 \pm 0.1$, $n = 49$ neurons; $p > 0.05$) or Rab11 (control RNAi, $2.4 \pm 0.3$, $n = 23$ neurons; RIM1 RNAi, $2.7 \pm 0.2$, $n = 16$ neurons; $p > 0.05$), indicating that the expression patterns of these proteins are unchanged (Supplementary Fig. 9a). Colocalization of GluN2B with Rab11 was significantly increased (control RNAi, $31 \pm 3\%$, $n = 23$ neurons; RIM1 RNAi, $39 \pm 2\%$, $n = 32$ neurons; $p < 0.05$; Fig. 8f), suggesting that RIM1 KD impairs the delivery of recycling GluN2B/NMDARs to the plasma membrane.

SNARE proteins have been implicated in the exocytosis of NMDAR vesicles at postsynaptic sites[8, 26]. In addition, it has been shown that RIM1 interacts directly or indirectly with the SNARE complex[37, 55]. We also assessed the interaction of RIM1 with SNAP25 in mouse cortex using coimmunoprecipitation (Supplementary Fig. 9b). To explore whether RIM1 is involved in

promoting the membrane fusion of recycling NMDARs, we knocked down RIM1 in cultured hippocampal neurons and assessed the colocalization of NMDARs with SNAP25. As shown in Supplementary Fig. 9a, RIM1 KD had no effect on the cluster density of SNAP25 (control RNAi, $3.3 \pm 0.3$, $n = 34$ neurons; RIM1 RNAi, $3.4 \pm 0.4$, $n = 18$ neurons; $p > 0.05$). However, RIM1 KD significantly decreased the colocalization of GluN2B with SNAP25 (GluN2B colocalized with SNAP25: control RNAi, $49 \pm 4\%$, $n = 15$ neurons; RIM1 RNAi, $37 \pm 4\%$, $n = 12$ neurons; $p < 0.05$; Fig. 8g). Collectively, these results suggest that RIM1 facilitates the fusion of recycling NMDARs with the surface membrane via acting as a Rab11 effector.

## Discussion

Previous studies have established a role for RIM1 in the release of synaptic vesicles at the presynaptic site. In the present study, we identified a postsynaptic role for RIM1 in mediating NMDAR trafficking and synaptic function (Supplementary Fig. 9c).

Evidence for RIM1 as an active zone molecule is based on the previous electron microscopic finding that RIM1 is exclusively localized presynaptically and the biochemical finding that RIM1 is one of the Rab3 effectors. Moreover, a presynaptic function of RIM1 has been confirmed across different synapses[29, 31, 33, 34, 36, 42, 56–58]. Using electron microscopic imaging and biochemical analysis, we confirmed that RIM1 was enriched in the presynaptic terminal in mouse hippocampus. However, we found that a small fraction of RIM1 was located postsynaptically. The difference between these electron microscopic data may be a result of the distinct brain regions surveyed, with our research focused on the mouse hippocampus while previous research focused on spinal motor neurons and the ribbon synapses of retinal photoreceptor cells[29, 31]. It is also possible that the postsynaptic pool is smaller and was neglected in previous work. In accord with our data, two identical proteomic analyses have identified RIM1 as a constituent of the PSD in the rat brain[59, 60]. Furthermore, we performed dual whole-cell recordings in hippocampal slices and showed that postsynaptic RIM1 determines the surface NMDAR level at SC synapses, but not that of AMPARs. Interestingly, previous work has reported that mice lacking RIM1 show impaired NMDAR-dependent late-phase LTP[61], but normal NMDAR-dependent early-phase LTP at hippocampal CA3–CA1 synapses[34], indicating that RIM1 is necessary for certain forms of NMDAR-dependent synaptic plasticity.

A numbers of active zone molecules involved in presynaptic vesicle release, including SNARE components and complexin[27, 62], have been shown to be located postsynaptically as well. However, most of these proteins mediate constitutive and/or activity-regulated AMPAR exocytosis, while only a few are involved in NMDAR exocytosis, including SNAP25, syntaxin-4, and VAMP1[13, 24]. Here, we identified RIM1 as another molecule involved in both presynaptic vesicle release and postsynaptic receptor trafficking, supporting the idea that postsynaptic receptor recycling shares mechanisms with presynaptic neurotransmitter release. We found that RIM1 specifically participates in NMDAR trafficking, but not AMPAR trafficking. RIM1 may provide a target for treating NMDAR-related pathological conditions, while leaving AMPAR-mediated basal synaptic transmission unaffected.

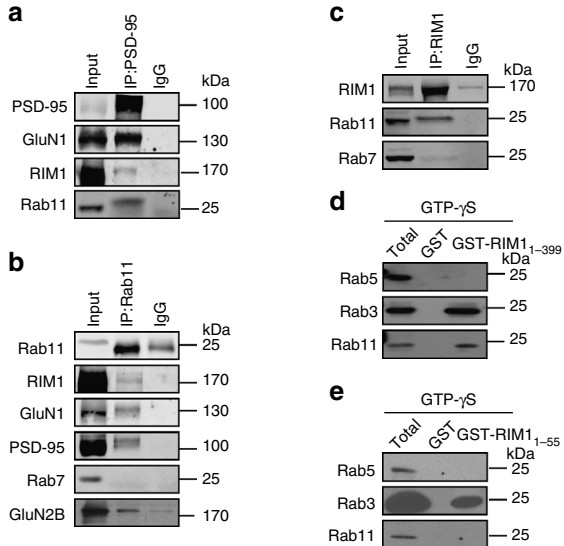

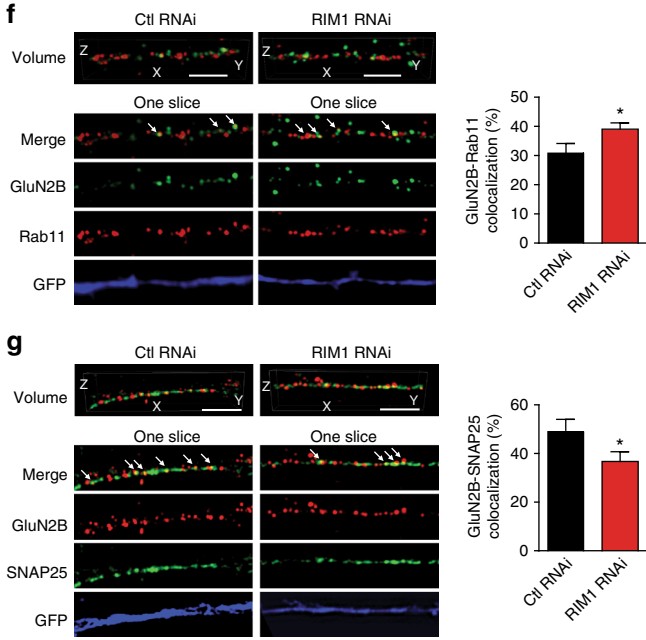

**Fig. 8** RIM1 acts as an effector of Rab11 and participates in the recycling of NMDARs. **a**–**c** Rab11 is involved in the RIM1/PSD-95/NMDAR complex in mouse cortex. Extracts of mouse cortex were immunoprecipitated with antibody against PSD-95, Rab11, or RIM1, and blotted with the corresponding antibodies. **d** GST-RIM1$_{1-399}$ interacts directly with Rab3 and Rab11, but not with Rab5, in the presence of GTP-γS. **e** GST-RIM1$_{1-55}$ has a direct interaction with Rab3, but not with Rab5 or Rab11, in the presence of GTP-γS. **f** RIM1 KD significantly increases the colocalization of Rab11 and GluN2B. Left: cultured hippocampal neurons were transfected with control RNAi ($n = 23$ neurons from three independent cultures) or RIM1 RNAi ($n = 32$ neurons from three independent cultures) plasmids at DIV6 and were then co-immunostained with antibodies against GluN2B and Rab11 at DIV10 and imaged by 3D-SIM. The lower four lines ("one slice") show one of the Z slices. Scale bar, 3 μm. Right: statistical analysis of colocalization of GluN2B and Rab11; $t$-test, *$p < 0.05$. **g** RIM1 KD significantly decreases the colocalization of SNAP25 and GluN2B. Left panel: cultured hippocampal neurons were transfected with control RNAi ($n = 15$ neurons from three independent cultures) or RIM1 RNAi ($n = 12$ neurons from three independent cultures) plasmids at DIV6. The neurons were immunostained with antibodies against GluN2B and SNAP25 at DIV10 and imaged by 3D-SIM. The lower four lines ("one slice") show one of the Z slices. Scale bar, 3 μm. Right panel: statistical analysis of colocalization of GluN2B and SNAP25; $t$-test, *$p < 0.05$

We found that RIM1 interacts directly with Rab11, indicating that postsynaptic RIM1, like presynaptic RIM1, binds to specific Rab proteins and modulates the trafficking of different vesicles that contain neurotransmitters or receptors. Notably, Rab11 and Rab3 are the only members of the Rab family involved in $Ca^{2+}$-induced exocytosis[63]. It will be interesting to determine the exact role of $Ca^{2+}$ in the modulation of NMDAR recycling.

Given that postsynaptic RIM1 is a key mediator of NMDAR trafficking, a critical question is whether postsynaptic RIM1 is linked to functional and behavioral phenotypes. In this study, we found that RIM1 KD in the CA1 region reduced NMDAR-mediated responses, confirming that postsynaptic RIM1 is involved in regulating synaptic function. Moreover, mice with RIM1 KD in the CA1 region had impaired hippocampus-dependent cognitive function and fear conditioning, but not perirhinal-dependent cognitive function, suggesting that postsynaptic RIM1 in the CA1 region plays a key role in hippocampus-dependent learning and memory. This is consistent with previous studies showing that RIM1$^{-/-}$ mice display cognitive deficits, since the presynaptic roles of RIM1 do not seem sufficient to cause such behavioral alterations[64]. RIM1 appears to be particularly important for a wide variety of classic schizophrenia-like behavioral abnormalities[65, 66], while NMDAR trafficking has been implicated in neurological diseases such as schizophrenia, autism, Alzheimer's disease, epilepsy, and chronic pain[67–70].

In conclusion, we have identified an RIM1-dependent mechanism that is specifically involved in NMDAR trafficking. This mechanism may participate in the fine-tuning of excitatory synapse functions in numerous physiological processes and neuropsychiatric disorders.

## Methods

**Animals.** Male C57BL/6 (3 or 8 weeks old) and RIM1$^{floxed}$ mice (3 weeks old)[41] were housed under a 12-h light/dark cycle with food and water provided ad libitum. All animal experiments were performed in accordance with the ethical guidelines of the Zhejiang University Animal Experimentation Committee and were in complete compliance with the National Institutes of Health Guide for the Care and Use of Laboratory Animals.

**Reagent.** Phenylmethanesulfonyl fluoride, aprotinin, phosphatase inhibitor cocktails, CNQX, (2R)-amino-5-phosphonovaleric acid (AP5), N-methyl-D-aspartic acid (NMDA), glutamate, picrotoxin, bicuculline, Forskolin, and Rolipram were from Sigma-Aldrich (St. Louis, USA). Tetrodotoxin (TTX) was from Tocris Bioscience (Ellisville, USA). Detailed information on the antibodies used is provided in the Supplementary Methods.

**Immunogold electron microscopy.** The pre-embedding immunogold labeling was carried out according to our previous method[71, 72]. In brief, mice were deeply anesthetized with sodium pentobarbital and transcardially perfused with 25 mL of 0.01 M phosphate buffered saline (PBS; pH 7.4) followed by 100 mL of 4% (w/v) paraformaldehyde and 0.05% glutaraldehyde in 0.1 M PB (pH 7.4). After perfusion, the brain was quickly removed and placed in the same fresh fixative without glutaraldehyde and post-fixed for an additional 2 h. The brain was then serially cut into 50-μm frontal sections on a microslicer. The sections were consecutively collected and placed in 0.1 M PB that contained 25% (w/v) sucrose and 10% (v/v) glycerol for 30 min for cryoprotection. Subsequently, the sections were freeze-thawed in liquid nitrogen to enhance antibody penetration during the immunohistochemical reaction. The sections were then placed in 0.05 M Tris-buffered saline (TBS, pH 7.4) containing 20% goat serum for 1 h to block non-specific binding. The sections were incubated for 24 h at room temperature with rabbit antiserum against RIM1 (1:500) in TBS, and then incubated with 1.4-nm gold particles conjugated to anti-rabbit IgG (1:50) for 12 h. The sections were then processed as follows: (1) post-fixation with 1% glutaraldehyde in 0.1 M PB for 10 min[50]; silver enhancement with an HQ Silver Kit (Nanoprobes, Stony Brook, NY, USA). The sections were then treated with 1% OsO$_4$ in 0.1 M PB for 1 h. Subsequently, the sections were counterstained with 1% (w/v) uranyl acetate in 70% ethanol for 1 h. After dehydration, the sections were mounted on silicon-coated glass slides and flat-embedded in epoxy resin (Durcupan; Fluka, Buchs, Switzerland).

Once the resin polymerized, the sections were examined under a light microscope and CA1 of the hippocampus was identified and excised. The tissue

samples of the selected regions were cut into 70-nm ultrathin sections with a diamond knife mounted on an ultramicrotome. The ultrathin sections were then mounted on single-slot grids coated with a Pioloform membrane, and stained with 1% (w/v) lead citrate. For each mouse, ~20 ultrathin sections beginning from the surface of the tissue block were collected and then examined in an electron microscope. RIM1-immunoreactive synaptic structures were imaged in 10–15 slices from each mouse. Synapses with typical presynaptic structures (with vesicles), postsynaptic structures (with PSDs), and a synaptic cleft were collected. RIM1-immunoreactive synapses were those with no less than 3 immunogold-silver grain particles distributed either pre- or postsynaptically. For each mouse, ~100 RIM1-immunoreactive synapses were randomly collected for statistics.

**Electrophysiology of cultured hippocampal neurons.** To record NMDA-evoked currents, whole-cell patch-clamp recordings were obtained from cultured hippocampal neurons transfected with RIM1-RNAi or control vector using the $Ca^{2+}$ phosphate method at DIV6 and recorded at DIV 9–10. During recordings, cells were bathed in an external solution containing (in mM): 129 NaCl, 5 KCl, 2 CaCl$_2$, and 10 glucose (pH 7.4), together with 1 μM tetrodotoxin and 50 μM bicuculline. Recording pipettes were filled with an intracellular solution containing (in mM): 135 CsMeSO$_4$, 8 NaCl, 10 HEPES, 4 MgATP, and 0.3 EGTA (pH 7.3). Recordings were performed at room temperature in voltage-clamp mode at a holding potential of −70 mV using a Multiclamp 700B amplifier (Molecular Devices, Sunnyvale, USA) and Clampex 10.2.0.12 software (Axon Instruments, Foster City, CA, USA). To activate NMDARs, we used 50 μM NMDA and 100 μM glycine for 2 s. Series resistance <20 MΩ was monitored for consistency during recordings. Non-NMDA currents were induced with 1 mM glutamate in the bath solution which contained 2 mM Mg$^{2+}$ and 10 μM AP5 together with 1 μM tetrodotoxin and 50 μM bicuculline. Cells with leak currents ≥300 pA were excluded from the analysis. The signals were amplified, sampled at 10 kHz, filtered to 3 kHz, and analyzed using Clampfit (Molecular Devices).

**In vivo injection and detection.** Three-week-old (8–12 g) and 8-week-old (22–25 g) mice were prepared for stereotaxic injection using standard procedures approved by the Zhejiang University Animal Experimentation Committee Panel on Laboratory Animal Care. Briefly, animals were anesthetized with pentobarbital (100 mg kg$^{-1}$ body weight) by intraperitoneal injection and then immobilized on a stereotaxic apparatus. A small volume of concentrated virus solution was injected into the CA1 region of the 3-week-old mice (300 nL, bregma = −1.8 mm; lateral 1.50 mm; ventral 1.50 mm), the CA3 region of the 3-week-old mice (500 nL, bregma = −1.6 mm; lateral 2.10 mm; ventral 2.20 mm), or the CA1 region of the 8-week-old mice (500 nL, bregma = 2.0 mm; lateral 1.6 mm; ventral 1.5 mm) with a microsyringe (World Precision Instruments Inc., Sarasota, FL, USA). The viral medium was injected into each hemisphere sequentially using a microinjection pump (Stoelting, Wood Dale, USA) at 0.1 μL min$^{-1}$. Ten to 15 days following the injection of virus, the animals were deeply anesthetized by intraperitoneal injection of pentobarbital (100 mg kg$^{-1}$ body weight) and then transcardially perfused with PBS (pH 7.4) followed by 4% paraformaldehyde in PBS. The brain was removed and post-fixed in 4% paraformaldehyde overnight at 4 °C. Coronal sections were cut at 50 μm on a Leica VT1200S vibratome (Leica Biosystems, Nussloch, Germany). Slides were finally coverslipped and mounted using ProLong Gold antifade reagent with or without DAPI (Invitrogen and Molecular Probes, Carlsbad, USA)

**Electrophysiological recording in slices and data analysis.** Ten to 15 days following the injection of virus, the animals were anesthetized with diethyl ether and the brain was rapidly removed and placed in ice-cold, high-sucrose cutting solution containing (in mM): 194 sucrose, 30 NaCl, 26 NaHCO$_3$, 10 glucose, 4.5 KCl, 1.2 NaH$_2$PO$_4$, 7 MgSO$_4$, 0.2 CaCl$_2$, and 2 MgCl$_2$. Slices were cut on a Leica vibratome in the high-sucrose cutting solution, and immediately transferred to an incubation chamber with artificial cerebrospinal fluid (ACSF) containing (in mM) 119 NaCl, 26.2 NaHCO$_3$, 11 glucose, 2.5 KCl, 1 NaH$_2$PO$_4$, 1.3 MgCl$_2$, 11 glucose, and 2.5 CaCl$_2$. The slices were allowed to recover at 34 °C for 30 min before being allowed to equilibrate at room temperature for another hour. During recordings, the slices were placed in a recording chamber constantly perfused with warmed ACSF (28–30 °C) and gassed continuously with 95% O$_2$ and 5% CO$_2$. All recordings were made with the GABA$_A$ receptor antagonist picrotoxin (100 μM) in the ACSF. Whole-cell recording pipettes (3–5 MΩ) were filled with a solution containing (in mM) 122.5 CsMeSO$_4$, 17.5 CsCl, 8 NaCl, 10 HEPES, 0.2 EGTA, 2 Mg$_2$ATP, 0.3 Na$_3$GTP, and 5 QX-314 (pH 7.25–7.3; osmolarity 290–299). Data were collected with a MultiClamp 700B amplifier and analyzed by pClamp10 software (Molecular Devices, Sunnyvale, USA). The initial access resistance was <25 MΩ, and was monitored throughout each experiment. Data were discarded if the access resistance changed >15% during an experiment. Data were filtered at 2 kHz (except for NMDAR-mEPSCs, which were filtered at 0.5 kHz), and digitized at 10 kHz.

A concentric bipolar stimulating electrode was placed in the stratum radiatum to evoke EPSCs in CA1 pyramidal cells. Cells were held at −70 mV to record AMPAR-mediated EPSCs and at +40 mV to record NMDAR-mediated EPSCs. The AMPAR/NMDAR ratio was calculated as the peak of the averaged AMPAR-mediated EPSC (30–50 consecutive events) at −70 mV divided by the averaged NMDAR-mediated EPSC (20–40 consecutive events) measured at 50 ms after the onset of the dual-component EPSC at +40 mV. NMDAR-mediated EPSCs were

recorded in the presence of CNQX (20 µM) and at a +40 mV holding potential. For comparison of the kinetics of the NMDAR-mediated EPSCs, 15–30 EPSCs were recorded, and the current decay was quantified as the time elapsed from 90% to 10% peak amplitude. The I–V relationship of NMDAR-mediated EPSCs was measured at holding potentials from −80 mV to +40 mV. Currents were normalized to peak responses at +40 mV. AMPAR-mEPSCs were recorded in the presence of TTX (1 µM) holding the cells at −70 mV. NMDAR-mEPSCs were recorded in the presence of TTX (1 µM) and CNQX (20 µM) and the cells were held at −70 mV in $Mg^{2+}$-free ACSF. Input–output curves were generated by evoking ten EPSCs every 0.1 Hz at pre-determined stimulation intensities. LTP was induced by two trains of high-frequency stimulation (100 Hz, 1 s) separated by 20 s, while cells were depolarized to 0 mV. This induction protocol was applied within 10 min of achieving the whole-cell configuration to avoid "wash-out" of LTP. The magnitude of LTP was calculated based on the EPSC values 35-45 min after the end of the induction protocol.

**Subcellular fractionation and Western blot analysis.** Subcellular fractionation was conducted with cortical tissue from adult C57BL/6 mice using an adapted protocol[19]. Briefly, the cortical samples were homogenized in 0.32 M sucrose buffer (10 mM sucrose and 10 mM HEPES, pH 7.4) to obtain the homogenate fraction, which was centrifuged ($1000 \times g$, 10 min, 4 °C) to obtain the S1 fraction. The S1 fraction was centrifuged (12,000 g, 20 min, 4 °C) to obtain the pellet (P2, crude synaptosome) fraction and the supernatant S2 fraction. To further digest the synaptosomes and yield an insoluble "PSD-enriched" membrane fraction and a "non-PSD-enriched" membrane fraction, we resuspended the P2 pellet in 4 mM HEPES buffer (4 mM HEPES and 1 mM EDTA, pH 7.4) and centrifuged again ($12,000 \times g$, 20 min, 4 °C). Resuspension and centrifugation were repeated. The resulting pellet was then resuspended with buffer A (20 mM HEPES, 100 mM NaCl, 0.5% Triton X-100, pH 7.2) and rotated slowly (15 min, 4 °C), followed by centrifugation ($12,000 \times g$, 20 min, 4 °C). The supernatant containing Triton X-100-soluble non-PSD membranes was retained. The pellet was resuspended in buffer B (20 mM HEPES, 0.15 mM NaCl, 1% TritonX-100, 1% deoxycholic acid, 1% sodium dodecyl sulfate, and 1 mM dithiothreitol, pH 7.5), followed by gentle rotation (1 h, 4 °C) and centrifugation ($10,000 \times g$, 15 min, 4 °C). The pellet was discarded, and the supernatant (Triton X-100-insoluble PSD fraction) was retained.

The fractionation of SPM and PSDs was prepared using an adapted protocol[40]. Mouse cortex was dissected in ice-cold PBS and then homogenized in 0.32 M HEPES-buffered sucrose solution with a Dounce homogenizer. The homogenate was centrifuged at $900 \times g$ for 10 min (4 °C) and the post-nuclear supernatant was further centrifuged at $10,000 \times g$ for 15 min (4 °C). The pellet (crude synaptosomal fraction, P2) was resuspended in 0.32 M HEPES-buffered sucrose and rotated for 30 min at 4 °C to ensure complete lysis. After centrifugation at $10,000 \times g$ for another 15 min, the pellet fraction (P2') was resuspended with 4-fold ddH₂O and homogenized with 3 strokes of a glass-Teflon homogenizer. Then, the solution was adjusted to 4 mM HEPES and rotated at 4 °C for 30 min. After centrifugation at $25,000 \times g$ for 20 min (4 °C), the pellet (synaptosomal membrane fraction) was resuspended with 0.32 M HEPES-buffered sucrose and laid on a discontinuous sucrose gradient containing 0.8 to 1.0 to 1.2 M sucrose. After another centrifugation at $150,000 \times g$ for 2 h (4 °C) using an SW41Ti rotor (Beckman), the SPM was collected from a cloudy band between 1.0 M and 1.2 M sucrose. The suspension was diluted to 0.32 M and further centrifuged at $200,000 \times g$ for 30 min at 4 °C using an MLA-150 rotor (Beckman). The pellet was resuspend in 50 mM HEPES/2 mM EDTA solution with 0.54% Triton X-100, rotated for 30 min at 4 °C, and centrifuged at $32,000 \times g$ for 20 min (4 °C) to obtain the Triton X-100-insoluble PSD fraction, which was then resuspended in 50 mM HEPES/2 mM EDTA.

Electrophoresis of equal amounts of total protein was placed on SDS-polyacrylamide gels, and the separated proteins were transferred onto nitrocellulose membranes (Hybond-C, GE Healthcare) at 4 °C. The membranes were blocked for 1 h at room temperature with 5% bovine serum albumin (BSA) in TBST (-buffered saline with Tween 20) and incubated with primary antibodies (5% BSA in TBST, 4 °C, overnight). After extensive washing, the membranes were incubated with the appropriate secondary antibody at 1:5000 for another hour. The unbound secondary antibodies were then washed, and the membranes were assessed on an Odyssey infrared imaging system (LI-COR). Or, the membranes were incubated with the appropriate HRP-coupled secondary antibody at 1:3000 for another hour, followed by enhanced chemiluminescence detection of the proteins with the Western Lightning Chemiluminescence Reagent Plus. All blots were repeated independently in triplicate and all the results shown were qualitatively consistent. Uncropped images of blots are shown in Supplementary Figs. 10 and 11.

**Statistical analysis.** No statistical method was used to predetermine sample size, but our sample sizes are similar to those reported in previous publications[26, 46, 62, 73]. For behavior and electrophysiological recording from brain slices, the mice with missed injections were excluded, and the investigator was blinded to the group allocation during the experiment. Data collection and processing were both randomized. Animals were selected randomly for all tests. Data are presented as the mean ± s.e.m. Data distribution was assumed to be normal but this was not formally tested. One-factor analysis of variance (ANOVA) was used to analyze data across multiple groups, Student's unpaired two-tailed t-test (referred

to as "t-test" throughout except as noted) or the paired t-test was used for all two-group comparisons, and the Kolmogorov–Smirnov test was used for cumulative probabilities. For all tests, $p < 0.05$ was considered statistically significant.

**Additional methods.** Details regarding the methods used for plasmid and antibodies, cell culture and transfection, coimmunoprecipitation of transfected HEK293T cells, immunostaining and surface biotinylation, imaging analysis, and behavioral testing are provided in the Supplementary Methods.

**Data availability.** All relevant data are available from the corresponding authors upon reasonable request.

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

## Acknowledgements

This work was supported by the National Natural Science Foundation of China (81471125, 81671049, and 91732102 to S.Q.; 81671162, 81521062, and 81561168022 to J. H.L.; 81671095 to T.C.; 91732105 to M.Z.; and 81371302 to W.Y.), the Zhejiang Science Fund for Distinguished Young Scholars (LR16C090001 to S.Q.), the National Basic Research Program of China (2013CB910204), Fundamental Research Funds for the Central Universities of China, the Chinese Ministry of Education Project 111 Program (B13026 to S.Q.), and Certificate of China Postdoctoral Science Foundation Grant (2018M630665 to Y.W.). M.Z. was also supported by the Ontario-China Research and Innovation Fund, the Azrieli Neurodevelopmental Research Program, and Brain Canada. We thank Dr. Jianyuan Sun (Chinese Academy of Sciences) and Dr. Thomas C. Südhof (Stanford University) for providing the RIM1^floxed and RIM2-knockout mice. We thank Dr. Ronald W. Holz (University of Michigan) and Dr. Weiping Zhang (Zhejiang University) for reagents. We thank Dr. Bong-Kiun Kaang (Seoul National University), Dr. Xiao-Dong Wang (Zhejiang University), and Dr. Xiangyao Li (Zhejiang University) for valuable suggestions on the manuscript. We also thank Guifeng Xiao and Shuangshuang Liu for technical support.

## Author contributions

J.W. performed the biochemical and immunostaining experiments; collected, analyzed, and interpreted data; and participated in writing the paper. X.Y.L. performed the electrophysiological and behavioral experiments; collected, analyzed, and interpreted data; and participated in writing the paper. X.L., R.Y., and T.C. performed the immunostaining and EM experiments; and collected, analyzed, and interpreted data. M.C. and M.J. performed immunohistochemical staining; Y.W., T.X., and W.F.D. performed the biochemical experiments; Y.Y. and C.C. performed molecular cloning experiments. M.Z. and W.Y. contributed intellectually and revised the manuscript. T.C., J.L., and S.Q. were responsible for the overall supervision of the study; designed the experiments; analyzed and interpreted data; and revised the paper.

## Additional information

**Competing interests:** The authors declare no competing interests.

