## [Peer Review File · Nature Communications]

Reviewers' expertise:

Reviewer #1: Glutamate receptors, LTP/LTD, learning and memory

Reviewer #2: Synaptic transmission, synaptic plasticity, presynaptic release

Reviewer #3: Synaptic cell biology, glutamate receptor trafficking

Additional reviewers:

Reviewer #4: Ultrastructural analysis of synapse

Reviewer #5: Ultrastructural analysis of synapse

Reviewers' comments:

Reviewer #1 (Remarks to the Author):

The manuscript by Wang et al reports a set of exciting and novel results that suggest a critical role of postsynaptic RIM1 in mediating recycling of NMDARs, and hence some hippocampus-dependent learning and memory. The manuscript addresses issues that are of general interest to neuroscientists at large. The results are largely of high quality and convincing. It merits consideration for publication in the journal provided that the following concerns are adequately addressed via a major revision.

Major concerns:

1. It remains unclear how RIM1 is recruited into the NMDAR complex. Using co-immunoprecipitation technique, the authors failed to demonstrate a direct interaction between RIM1 and NMDARs. However, they did not include a positive control to show their success in co-immunoprecipitation of a partner protein in their experiments. Is Rab11 the adaptor protein to link RIM1 to NMDAR complex? If so, can authors show the ability of Rab11 N-terminal peptide to remove RIM1 from the NMDAR complex in a co-immunoprecipitation experiment?

2. Since RIM1 KD significantly reduces the surface expression of NMDARs at the synapse, one would expect that it should also have a significant impact on NMDAR-dependent synaptic plasticity such as LTP and/or LTD. Authors should examine this experimentally and thereby provide a better link between electrophysiological changes to the behavioural alterations they observed.

Other points:

1. Fig. 4b and c: a positive control is needed for authors to demonstrate their success in immunoprecipitating an interacting protein in their recombinant expression systems; otherwise one cannot be sure that the negative co-immunoprecipitation is a result of no direct interaction between RIM1 and NMDARs or a technical failure in co-immunoprecipitation.

2. Fig. 5b: the authors should also show no change in AMPA-induced currents in the same cells.

3. Fig. S2: Given that RIM1 KD impairs the recycling of NMDARs to the plasma membrane, thereby directing more of them into lysosomes for degradation, why was no overall change of total NMDARs observed?

4. Fig. S3: Why was there no significant reduction in the amplitude of NMDAR currents by RIM RNAi? This is not consistent with the results shown in Fig. 2F and G, and Fig. 5b.

5. Fig. S6: The membranes of these blots are required to be stripped and re-probed with a membrane protein to ensure equal biotinylation efficacy and similar loading of biotinylated proteins. Additionally, the normalization should be done with this internal membrane protein. GluA1 or A2 would be highly recommended as RIM did not affect their surface expression.

Reviewer #2 (Remarks to the Author):

In this study, the authors propose that RIM, a key component of the presynaptic active zone, also operates on the postsynaptic side organizing NMDA receptors. While this is a potentially interesting and a rather provocative proposal, the data provided in the manuscript does not make a convincing case. The authors need to upgrade this work substantially to bolster their case.

1. Subcellular fractionation result could simply be due to contamination as RIM is a large sticky protein. The authors should control if they detect other active zone proteins under the same conditions (munc-13, Bassoon, Piccolo etc.)

2. NMDA response analysis should include NMDA-mEPSC properties (in particular amplitude distributions). This is a key piece to provide information about unitary response properties (per synapse basis).

3. The results should be reconciled with earlier work. Parallel use of RIM1KOs would be one option. Especially this setting would allow the authors check the specificity of their knock down approach. In earlier work, NMDA receptor mediated responses were relatively unaffected by the loss of RIM1 (e.g. MK-801 block experiments in Schoch et al., 2002 Nature)

4. The knock down approach needs validation with rescue experiments as well as possibly presynaptic knock down to reproduce earlier work.

5. If the decrease in NMDA responses is as robust as the authors propose, one would expect to see some degree of deficits in CA3-CA1 LTP in RIM1 KOs.

Reviewer #3 (Remarks to the Author):

This work indicates that Rim1, better known as a presynaptic structural protein, functionally interacts with the NMDAR to foster its surface expression. The results are generally of interest, the experiments largely executed very well except sufficient control for off target effects of the RIM1 KD is lacking in several experiments (see below)

Major Concern

1. Figure 1A: immunoreactivity for RIM1 and RIM2 in the PSD fraction is minimal and could be non-specific; the Supplemental Fig. 1a shows much stronger signals in the PSD fraction there – how reproducible is the detection of RIM1 and RIM2 in the PSD fraction? Some sort of quantification should be provided over several independent experiments/PSD preparations or images from several PSD preparations should be shown. Figure 1B/C: the EM labeling for RIM is not convincing – the labeling and image quality is low. If the authors want to use such imaging they need to show that the label is really specific by showing its absence in RIM KO tissue. If this is not possible the labeling could be taken out of the manuscript if the PSD fractionation issue can be thoroughly addressed.

2. RIM1 KD could have off target effects. It is unclear whether the control RNAi forms a hairpin loop and truly mimics function RNAi (a more detailed description is needed here). Even if so, the authors should use a rescue strategy in selected experiments in Figures 2 (EPSC of NMDARs as in panel h) and Figure 3 (novel object learning; e.g., panels a-c or d/e), analogous to the rescue in Figure 5a.

3. Monospecificity of the N-terminal GluN2B antibody as to be demonstrated in neurons. To show specificity in HEK cells ectopically expressing GluN2B as in Figures S6a is not sufficient. Ideally the authors would show that surface staining for GluN2B is absent in neurons after GluN2B knock down or knock out. If this is not feasible at least they should show monospecificity by immunoblotting tissue from wild type and KO animals with a higher resolution in the 200 kDa range than the one in Figure S6a.

4. It is unclear whether the authors cleared their tissue extracts by ultracentrifugation (at least 100,000 x g for 1 h) to remove non-solubilized material before immunoprecipitations. If not, key experiments have to be repeated after ultracentrifugation. Otherwise any co-immunoprecipitation could be due to colocalization of the NMDAR and the other proteins within same secretory vesicles rather than within a bona fide protein complex.

Minor Concern

5. It would be desirable to show that the GluN2B surface labeled puncta are mostly synaptic (co-localization with a synaptic marker) to instill confidence in this labeling.

Reviewers' comments:

Reviewer #1 (Remarks to the Author):

The manuscript by Wang et al reports a set of exciting and novel results that suggest a critical role of postsynaptic RIM1 in mediating recycling of NMDARs, and hence some hippocampus-dependent learning and memory. The manuscript addresses issues that are of general interest to neuroscientists at large. The results are largely of high quality and convincing. It merits consideration for publication in the journal provided that the following concerns are adequately addressed via a major revision.

Major concerns:

1. It remains unclear how RIM1 is recruited into the NMDAR complex. Using co-immunoprecipitation technique, the authors failed to demonstrate a direct interaction between RIM1 and NMDARs. However, they did not include a positive control to show their success in co-immunoprecipitation of a partner protein in their experiments. Is Rab11 the adaptor protein to link RIM1 to NMDAR complex? If so, can authors show the ability of Rab11 N-terminal peptide to remove RIM1 from the NMDAR complex in a co-immunoprecipitation experiment?

Responses: Thanks to the reviewer's valuable comments.

- (1) We agree with the reviewer that we didn't describe how RIM1 is recruited into the NMDAR complex in our previous manuscript. To address this issue, we designed a series of experiments in our revised manuscript. Given that PSD95 is a critical scaffolding protein within the NMDAR complex, as well as our finding that RIM1 and PSD95 are within the same protein complex (Figure 4a in the previous manuscript), we tested whether PSD95 is involved in coupling RIM1 with NMDARs. We co-transfected mCherry-PSD95 with HA-RIM1 or with GFP-GluN2B into the HEK293T cells and examined the interaction of PSD95 with RIM1 or GluN2B using co-immunoprecipitation. In accordance with previous finding, PSD95 interacted with GluN2B in the HEK293T cells (Figure 5e in the revised manuscript). Additionally, we observed the interaction between PSD95 and RIM1 (Figure 5d in the revised manuscript). More importantly, when mCherry-PSD95, HA-RIM1, and GFP-GluN2B were triple transfected into the HEK293T cells, we observed the interaction between GluN2B and RIM1 (Figure 5f in the revised manuscript). We further demonstrated that the C terminus of RIM1 (RIM1₇₀₅₋₁₆₁₅) mediated the interaction between RIM1 and PSD95 (Figure 5g in the revised manuscript). Combined together, these results suggest that PSD95 may be the linker that couples RIM1 with NMDARs. We have added these new data in our revised manuscript (Figure 5d-5g) and also modified our working model in Figure S7.
- (2) In our revised manuscript, we also tried to screen the domain mediating the interaction between RIM1 and Rab11. We focused on the N-terminal alpha helix (RIM1₁₋₅₅) that was found to interact with Rab3 (Wang et al., 2001). Unfortunately, no direct interaction was observed between RIM1₁₋₅₅ and Rab11 (Figure 8e in the revised manuscript). Therefore, based on our current finding, it is still difficult to design peptides to block the interaction of RIM1 with Rab11.

2. Since RIM1 KD significantly reduces the surface expression of NMDARs at the synapse, one would expect that it should also have a significant impact on NMDAR-dependent synaptic plasticity such as LTP and/or LTD. Authors should examine this experimentally and thereby provide a better link between electrophysiological changes to the behavioural alterations they observed.

Responses: Thanks to the reviewer's valuable suggestion.

In the revised manuscript, we analyzed NMDAR-dependent LTP in the hippocampal CA1 region and observed that NMDAR-dependent LTP was impaired when RIM1 was knocked down in area CA1 (Figure 2f in the revised manuscript). This result is in accordance with our previous electrophysiological and behavioral findings.

Other points:

1. Fig. 4b and c: a positive control is needed for authors to demonstrate their success in immunoprecipitating an interacting protein in their recombinant expression systems; otherwise one cannot be sure that the negative co-immunoprecipitation is a result of no direct interaction between RIM1 and NMDARs or a technical failure in co-immunoprecipitation.

Responses: Thanks to the reviewer's suggestion.

As we mentioned in our upper response to major point #1, we have co-transfected PSD95 with GluN2B or with RIM1 into the HEK293T cells and observed their interaction using co-immunoprecipitation (Figure 5d and 5e in the revised manuscript). These data demonstrate our success in immunoprecipitation in the recombinant expression systems

2. Fig. 5b: the authors should also show no change in AMPA-induced currents in the same cells.

Responses: Thanks to the reviewer's suggestion.

In the revised manuscript, we analyzed the non-NMDA currents in the cultured hippocampal neurons with or without RIM1 KD and found that non-NMDA currents in the RIM1 KD neurons showed no difference from that of the Ctl neurons (Figure S6d)

3. Fig. S2: Given that RIM1 KD impairs the recycling of NMDARs to the plasma membrane, thereby directing more of them into lysosomes for degradation, why was no overall change of total NMDARs observed?

Responses: Thanks to the reviewer's comments.

In our manuscript, we have observed that RIM1 KD increased the colocalization of NMDARs with Rab11, but decreased the colocalization of NMDARs with SNAP25 (Figure 8f and 8g in the revised manuscript). It suggests that NMDAR-containing vesicles cannot be fused with the surface membrane and are retained in the recycling endosomes.

4. Fig. S3: Why was there no significant reduction in the amplitude of NMDAR currents by RIM RNAi? This is not consistent with the results shown in Fig. 2F and G, and Fig. 5b.

Responses: Thanks to the reviewer's comments.

I/v relationships were designed to detect the electrical feature of the NMDA receptors. In this experiment, RIM1 KD neurons and ctl neurons were recorded in the different slices and were evoked by different stimulus intensities. The data shown in Figure 2e is normalized. That is why there was no significant reduction in the amplitude of NMDAR currents in RIM1 KD neurons.

5. Fig. S6: The membranes of these blots are required to be stripped and re-probed with a membrane protein to ensure equal biotinylation efficacy and similar loading of biotinylated proteins. Additionally, the normalization should be done with this internal membrane protein. GluA1 or A2 would be highly recommended as RIM did not affect their surface expression.

Responses: Thanks to the reviewer's valuable suggestion.

We have designed the experiment according to the reviewer's suggestion, such as stripping the membrane of the GluN1 blots, re-probing it with cadherin, and normalizing it with cadherin. We still observed significant decrease of surface GluN1 in RIM1 KD group compared with the control group (Figure S6e in the revised manuscript).

It should be mentioned that we also observed the decrease of surface GluA1 by using biotinylation (Figure 1 in the rebuttal letter). In contrast, we didn't detect the change of AMPARs by using surface staining (Figure S6c) or electrophysiological recording (Figure S6d). This difference is possibly due to the different transfection methods we used in these experiments. We infected the cultured neurons with virus to knock down RIM1 efficiently for the biotinylation assay, whereas we used Ca²⁺-phosphate transfection method with plasmids for surface staining and electrophysiological recording. As we mentioned in our manuscript, Ca²⁺-phosphate transfection method with the transfection efficiency around 5% (Washbourne and McAllister, 2002; Xia et al., 1996) was more suitable for our studies since it minimizes the interference of presynaptic effects. Virus infection with much higher transfection efficiency may lead to both presynaptic and postsynaptic impairments. It is possible that presynaptic RIM1 KD *in vitro* has effects on the surface expression of postsynaptic AMPARs.

Figure 1 Cultured hippocampal neurons infected with RIM1 RNAi lentivirus have lower surface expression of GluA1 compared to neurons infected with Ctl RNAi lentivirus. *p < 0.05, n = 3.

Reviewer #2 (Remarks to the Author):

In this study, the authors propose that RIM, a key component of the presynaptic active zone, also operates on the postsynaptic side organizing NMDA receptors. While this is a potentially interesting and a rather provocative proposal, the data provided in the manuscript does not make a convincing case. The authors need to upgrade this work substantially to bolster their case.

1. Subcellular fractionation result could simply be due to contamination as RIM is a large sticky protein. The authors should control if they detect other active zone proteins under the same conditions (munc-13, Bassoon, Piccolo etc.)

Responses: Thanks to the reviewer's valuable suggestion.

We used Rab3, Synaptophysin and synapsin as the presynaptic marker in our previous manuscript. In the revised manuscript, we added Bassoon as another presynaptic marker, and found that it was completely located in the presynaptic fraction, but not in the postsynaptic fraction (Figure S1a in the revised manuscript).

2. NMDA response analysis should include NMDA-mEPSC properties (in particular amplitude distributions). This is a key piece to provide information about unitary response properties (per synapse basis).

Responses: Thanks to the reviewer's valuable suggestion.

We have analyzed NMDA-mEPSC properties in the revised manuscript. As shown in Figure S3d, the amplitude of RIM1 KD group showed a significant decrease compared with that of the control group. The mean inter-event intervals were not influenced, while the cumulative distributions showed slight rightward distribution shifts ($p = 0.0426$). It is possible that the events are more likely to be ignored in the KD group.

3. The results should be reconciled with earlier work. Parallel use of RIM1KOs would be one option. Especially this setting would allow the authors check the specificity of their knock down approach. In earlier work, NMDA receptor mediated responses were relatively unaffected by the loss of RIM1 (e.g. MK-801 block experiments in Schoch et al., 2002 Nature)

Responses: Thanks to the reviewer's comments.

To examine whether our results are reconciled with earlier work, we designed an experiment to specifically knock down presynaptic RIM1 and test its role in neurotransmitter release. We injected RIM1 KD virus in the CA3 region and recorded PPF in the CA1 region. As shown in Figure S3b and S3c, RIM1 KD in the CA3 region affected PPF in the CA1 region, in accordance with the previous finding that RIM1 is critical in neurotransmitter release (Schoch et al., 2002).

MK-801 block experiment in Schoch et al., 2002 nature is designed to detect the Pr (Hessler et al., 1993; Rosenmund et al., 1993) and do not reflect the properties of NMDA receptor-mediated responses.

4. The knock down approach needs validation with rescue experiments as well as possibly presynaptic knock down to reproduce earlier work.

Responses: Thanks to the reviewer's comment.

(1) In our previous manuscript, we have used RIM1 rescue plasmid to confirm that RIM1 KD specifically influences surface GluN2B localization (Figure 6a in the original manuscript). In

the revised manuscript, we added another group of electrophysiological recording to show that RIM1 rescue plasmid can reverse the effect of RIM1 KD on NMDAR currents (Figure 6b in the revised manuscript).

- (2) As we mentioned in our response to point #3, we designed a new experiment to specifically knock down presynaptic RIM1 and test its role in neurotransmitter release in the revised manuscript. We injected RIM1 KD virus in the CA3 region and recorded PPF in the CA1 region, and observed that presynaptic RIM1 KD affects PPF in the CA1 region (Figure S3b and S3c), in accordance with previous finding that RIM1 is critical in neurotransmitter release (Schoch et al., 2002).

5. If the decrease in NMDA responses is as robust as the authors propose, one would expect to see some degree of deficits in CA3-CA1 LTP in RIM1 KOs.

Responses: Thanks to the reviewer's comments.

In the revised manuscript, we analyzed NMDAR-dependent LTP in the hippocampal CA1 region. As shown in Figure 2f, NMDAR-dependent LTP was significantly impaired when RIM1 was knocked down in the CA1 region.

Reviewer #3 (Remarks to the Author):

This work indicates that Rim1, better known as a presynaptic structural protein, functionally interacts with the NMDAR to foster its surface expression. The results are generally of interest, the experiments largely executed very well except sufficient control for off target effects of the RIM1 KD is lacking in several experiments (see below)

Major Concern

1. Figure 1A: immunoreactivity for RIM1 and RIM2 in the PSD fraction is minimal and could be non-specific; the Supplemental Fig. 1a shows much stronger signals in the PSD fraction there – how reproducible is the detection of RIM1 and RIM2 in the PSD fraction? Some sort of quantification should be provided over several independent experiments/PSD preparations or images from several PSD preparations should be shown. Figure 1B/C: the EM labeling for RIM is not convincing – the labeling and image quality is low. If the authors want to use such imaging they need to show that the label is really specific by showing its absence in RIM KO tissue. If this is not possible the labeling could be taken out of the manuscript if the PSD fractionation issue can be thoroughly addressed.

Responses: Thanks to the reviewer's comment.

- (1) In our original manuscript, we used different protocols to get the PSD fraction, such as protocols based on Triton solubility in Figure 1a and on sucrose gradient in Figure 1b. In the revised manuscript, we compared the concentration of RIM1, as well as other synaptic markers, using one fold or two fold quantity of PSD fraction sample (Figure S1a). The results showed that the abundance of RIM1 and PSD95 were significantly increased in the two fold

quantity of PSD fraction sample, while synaptophysin and the active zone protein bassoon were still undetectable in the PSD fraction.

- (2) We apologized for the quality of the EM images. In the revised manuscript, we repeated the immunogold staining and replaced the old images with better one (Figure 1c).
- (3) To test the specificity of the antibody against RIM1, we injected AAV-hSyn-Cre-EGFP into the hippocampus of Rim1^{flox^{ed}} mice. 3 weeks after injection, the mouse hippocampus was sectioned and stained with antibody against Rim1. As shown in Figure S1b, GFP positive cells showed much less RIM1 fluorescent intensity compared to the GFP negative cells.

2. RIM1 KD could have off target effects. It is unclear whether the control RNAi forms a hairpin loop and truly mimics function RNAi (a more detailed description is needed here). Even if so, the authors should use a rescue strategy in selected experiments in Figures 2 (EPSC of NMDARs as in panel h) and Figure 3 (novel object learning; e.g., panels a-c or d/e), analogous to the rescue in Figure 5a.

Response: Thanks to the reviewer's comment.

- (1) The nonspecific scramble shRNA used in our manuscript (5'-TTCTCCGAACGTGTACAGT-3') was purchased from GeneChem (Shanghai, China). This sequence can form a hairpin loop and mimic functional RNAi and has been used as negative control of RNAi (Adorno et al., 2013; Cordenonsi et al., 2011). We have added this information in the revised manuscript (*Plasmids, drugs, and antibodies* sections of Methods in the revised manuscript, p24, l12).
- (2) In the revised manuscript, we designed another rescue experiment to transfect the RIM1 rescue plasmid into the cultured cortical neurons and detect its effect on NMDAR currents. As shown in Figure 6b in the revised manuscript, it completely reversed RIM1 KD-induced reduction of NMDAR currents.
- (3) The rescue strategy in our original Figure 5a is that we transfected RIM1 rescue plasmid (RIM1 mRNA with the sequence "AGAATGGACCACAAATGCTT" mutated to "AGGATGGATCATAAGTGTTT" was inserted into the GFP-tagged RIM1 shRNA plasmid) into the cultured neurons. To do such kind of rescue experiments in Figure 2 or Figure 3, we have to use virus infection. However, the full length of RIM1 mRNA is 4842bp, and it is technically difficult to package the virus. We have asked two companies in China, the Genechem (Shanghai, China) and Obio Technology (Shanghai, China) to package Lentivirus or AAV for our RIM1 rescue experiments. Unfortunately, Genechem refused to do it for high risk, while Obio Technology tried three times, but the quality of the vector is poor and we cannot observe live neurons with GFP fluorescence after injection in the CA1 region. We have also contacted with Dr. Chen, the associate director of Penn Vector Core Gene Therapy Program, and he responded to us as "Please keep in mind that the packaging capacity of AAV is about 4.9 Kd, from ITR to ITR and including 2 ITRs. We can try to package AAV genome exceeding 4.9 kd, however, we cannot guarantee the yield and the quality of the vector". Therefore, we appreciate the reviewer for this suggestion, but now we cannot do such kind of rescue experiment.

3. Monospecificity of the N-terminal GluN2B antibody as to be demonstrated in neurons. To show specificity in HEK cells ectopically expressing GluN2B as in Figures S6a is not sufficient. Ideally the

authors would show that surface staining for GluN2B is absent in neurons after GluN2B knock down or knock out. If this is not feasible at least they should show monospecificity by immunoblotting tissue from wild type and KO animals with a higher resolution in the 200 kDa range than the one in Figure S6a.

Responses: Thanks to the reviewer's suggestion.

In the revised manuscript, we knocked down GluN2B in the cultured neuron and detected the surface localization of GluN2B in these neurons. As shown in Figure S5c, surface GluN2B is significantly decreased in GluN2B KD neurons.

4. It is unclear whether the authors cleared their tissue extracts by ultracentrifugation (at least 100,000 x g for 1 h) to remove non-solubilized material before immunoprecipitations. If not, key experiments have to be repeated after ultracentrifugation. Otherwise any co-immunoprecipitation could be due to colocalization of the NMDAR and the other proteins within same secretory vesicles rather than within a bona fide protein complex.

Responses: Thanks to the reviewer's comments.

As we mentioned in our previous manuscript, tissue coimmunoprecipitation was carried out under nondenaturing conditions for membrane solubilization (Figure 5a; Figure 8a-8c) (Luo et al., 1997). Briefly, the supernatant of tissue homogenate was added with 0.10 vol of 10% sodium deoxycholate and then 0.10 vol of 1% Triton X-100, and the preparation was dialyzed against binding buffer (50 mM Tris-HCl, pH 7.4, 0.1% Triton) overnight at 4°C. After centrifugation (37,000g, 4°C) for 15 min, IP antibody was added to the supernatant and incubated overnight at 4°C. This protocol has been proved to avoid artificial stickiness (Luo et al., 1997) and has been used in a lot of labs to detect the interaction of membrane proteins (Al-Hallaq et al., 2007; Fukata et al., 2005; McQuail et al., 2016; Oku and Haganir, 2013).

HEK293T cells for coimmunoprecipitation was lysed by RIPA lysis buffer (50 mM Tris, pH7.4, 150mM NaCl, 1% NP-40, 0.25% sodium deoxycholate) and then centrifuged (20,000g, 4°C) for 15 min (Figure 5b-5g). The detergents, such as NP-40 and sodium deoxycholate, solubilize membranes, and thus co-IP of the NMDAR and RIM1 could not be within the same secretory vesicles. Moreover, we didn't observe the interaction between RIM1 and NMDARs (Figure 5c) in the transfected HEK293T cells, unless PSD95 was cotransfected with them (Figure 5f), further indicating that vesicle membrane is not enough to bring these proteins into complexes.

Minor Concern

5. It would be desirable to show that the GluN2B surface labeled puncta are mostly synaptic (co-localization with a synaptic marker) to instill confidence in this labeling.

Responses: Thanks to the reviewer's suggestion.

In the revised manuscript, we detected the colocalization of surface GluN2B with PSD95. As shown in Figure S5b, surface-labeled GluN2B was partially colocalized with PSD95.

Reference

- Adorno, M., Sikandar, S., Mitra, S.S., Kuo, A., Nicolis Di Robilant, B., Haro-Acosta, V., Ouadah, Y., Quarta, M., Rodriguez, J., Qian, D., *et al.* (2013). Usp16 contributes to somatic stem-cell defects in Down's syndrome. *Nature* *501*, 380-384.
- Al-Hallaq, R.A., Conrads, T.P., Veenstra, T.D., and Wenthold, R.J. (2007). NMDA di-heteromeric receptor populations and associated proteins in rat hippocampus. *The Journal of neuroscience : the official journal of the Society for Neuroscience* *27*, 8334-8343.
- Cordenonsi, M., Zanconato, F., Azzolin, L., Forcato, M., Rosato, A., Frasson, C., Inui, M., Montagner, M., Parenti, A.R., Poletti, A., *et al.* (2011). The Hippo transducer TAZ confers cancer stem cell-related traits on breast cancer cells. *Cell* *147*, 759-772.
- Fukata, Y., Tzingounis, A.V., Trinidad, J.C., Fukata, M., Burlingame, A.L., Nicoll, R.A., and Brecht, D.S. (2005). Molecular constituents of neuronal AMPA receptors. *The Journal of cell biology* *169*, 399-404.
- Hessler, N.A., Shirke, A.M., and Malinow, R. (1993). The probability of transmitter release at a mammalian central synapse. *Nature* *366*, 569-572.
- Luo, J., Wang, Y., Yasuda, R.P., Dunah, A.W., and Wolfe, B.B. (1997). The majority of N-methyl-D-aspartate receptor complexes in adult rat cerebral cortex contain at least three different subunits (NR1/NR2A/NR2B). *Molecular pharmacology* *51*, 79-86.
- McQuail, J.A., Beas, B.S., Kelly, K.B., Simpson, K.L., Frazier, C.J., Setlow, B., and Bizon, J.L. (2016). NR2A-Containing NMDARs in the Prefrontal Cortex Are Required for Working Memory and Associated with Age-Related Cognitive Decline. *The Journal of neuroscience : the official journal of the Society for Neuroscience* *36*, 12537-12548.
- Oku, Y., and Haganir, R.L. (2013). AGAP3 and Arf6 regulate trafficking of AMPA receptors and synaptic plasticity. *The Journal of neuroscience : the official journal of the Society for Neuroscience* *33*, 12586-12598.
- Rosenmund, C., Clements, J.D., and Westbrook, G.L. (1993). Nonuniform probability of glutamate release at a hippocampal synapse. *Science* *262*, 754-757.
- Schoch, S., Castillo, P.E., Jo, T., Mukherjee, K., Geppert, M., Wang, Y., Schmitz, F., Malenka, R.C., and Sudhof, T.C. (2002). RIM1alpha forms a protein scaffold for regulating neurotransmitter release at the active zone. *Nature* *415*, 321-326.
- Wang, X., Hu, B., Zimmermann, B., and Kilimann, M.W. (2001). Rim1 and rabphilin-3 bind Rab3-GTP by composite determinants partially related through N-terminal alpha-helix motifs. *The Journal of biological chemistry* *276*, 32480-32488.
- Washbourne, P., and McAllister, A.K. (2002). Techniques for gene transfer into neurons. *Current opinion in neurobiology* *12*, 566-573.
- Xia, Z., Dudek, H., Miranti, C.K., and Greenberg, M.E. (1996). Calcium influx via the NMDA receptor induces immediate early gene transcription by a MAP kinase/ERK-dependent mechanism. *The Journal of neuroscience : the official journal of the Society for Neuroscience* *16*, 5425-5436.

Reviewers' comments:

Reviewer #1 (Remarks to the Author):

Authors have adequately addressed all concerns raised in the first round of reviewing process, and it can now be accepted for the publication in the journal.

Reviewer #2 (Remarks to the Author):

The authors have addressed my earlier questions. The findings are rather provocative. I am not sure if one can reconcile these findings with the enormous existing literature on RIM1. Therefore, this manuscript will trigger a debate, which will likely be resolved with further independent studies.

Reviewer #3 (Remarks to the Author):

The authors satisfactorily addressed a number of concerns but critical issues remain and need to be further addressed.

Major Concern

1. Figure 1A: I maintain that immunoreactivity for RIM1 and RIM2 in the PSD fraction is minimal and could be non-specific; as I stated before the Supplemental Fig. 1a shows much stronger signals in the PSD fraction there – the authors need to address the question how reproducible the detection of RIM1 and RIM2 in the PSD fraction really is, e.g. quantification should be provided over several independent experiments/PSD preparations or images from several PSD preparations should be shown for all the fractions and several marker proteins. A new concern is that the authors do not detect bassoon in their PSD preparation. Although bassoon is widely considered to be only present at the presynaptic site, many different studies using the same method for PSD preparations as the one the authors (based on Carlin et al., 1980) clearly find substantial amounts of bassoon in their PSD fractions (e.g., tom Dieck et al., 1998: J Cell Biol 142, 499-509; Walikonis et al., 2000: J Neurosci 20, 4069-4080; Yoshiyuki et al., 2004: J Neurochem 88, 759-768; Lowenthal et al., 2015: J Proteome Res 14, 2528-2538). Although the presence of bassoon is considered to be a contamination, it is highly reproducible and a specific localization at the PSD cannot be ruled out. That the authors no report a lack of bassoon raises some concern of how their PSD preparation compares to those by many others.

2. Figure 1C: the EM labeling for RIM remains to be unconvincing. Fig1 C left is supposedly showing presynaptic labeling for RIM but the label is not very strong and could be due to variability in obtaining contrast. Even worse, the opposing structure is anything but a typical postsynaptic site-it seems a very wide and elongated structure with mitochondria and/or other large organelles –this is certainly not a postsynaptic site of a glutamatergic synapse, which is typically a dendritic spine. In the middle panel the authors identify a dendritic spine

with apparently strong immunostaining. However, none of the presynaptic sites in this image are, curiously enough, labeled – this raises serious concerns that the labeling in these images is random and not specific because most if not all presynaptic sites should have RIM.

3. Solubilization of NMDARs is not trivial. And centrifugation at 37,000 x g will not bring all membrane fragments into the pellet-one can look this up in any handbook on subcellular fractionation. Now there has been evidence that 1% Triton X-10 solubilizes ~ 85% of NMDARs in the ER-derived microsomal fraction (Blahos and Wenthold, 1996: J Biol Chem 271, 15669-74) and the addition of 0.5% DOC might solubilize most of the remainder, which would minimize the potential for co-IP of RIM simply by association with the same membrane fragments rather than direct binding to the NMDAR complex. However, that all membranes have been solubilized has to be proven by the authors in order to make this claim and that proof requires sedimentation of non-solubilized material by ultracentrifugation. It would be easier for the authors to repeat 1-2 coIP experiments and apply ultracentrifugation at 100,000 x g for 1 h to eliminate this issue. In the absence of either evidence the conclusion that RIM and NMDARs are in the same protein complexes is not well enough supported.

Reviewer #1 (Remarks to the Author):

Authors have adequately addressed all concerns raised in the first round of reviewing process, and it can now be accepted for the publication in the journal.

We thank the reviewer for positive comments and are happy that the reviewer agrees that our work is suitable for the publication in Nature Communications.

Reviewer #2 (Remarks to the Author):

The authors have addressed my earlier questions. The findings are rather provocative. I am not sure if one can reconcile these findings with the enormous existing literature on RIM1. Therefore, this manuscript will trigger a debate, which will likely be resolved with further independent studies.

We thank the reviewer for very thoughtful and positive comments. Indeed, we also anticipate that our work will be likely of great interest to people who cares about synaptic function, organization and plasticity. As the first group to show the role of postsynaptic RIM1, we are excited and honored to see how our work will be followed and validated by other groups, which hopefully will ultimately clarify our understanding of RIM1, a key synaptic protein that we all care.

Reviewer #3 (Remarks to the Author): The authors satisfactorily addressed a number of concerns but critical issues remain and need to be further addressed.

We appreciate that the reviewer thinks we have satisfactorily addressed some of his concerns and are still interested to see how some other issues he gets bothered to be further addressed.

Major Concern

1. Figure 1A: I maintain that immunoreactivity for RIM1 and RIM2 in the PSD fraction is minimal and could be non-specific; as I stated before the Supplemental Fig. 1a

shows much stronger signals in the PSD fraction there – the authors need to address the question how reproducible the detection of RIM1 and RIM2 in the PSD fraction really is, e.g. quantification should be provided over several independent experiments/PSD preparations or images from several PSD preparations should be shown for all the fractions and several marker proteins.

We thank the reviewer for bringing this out again. If we understand correctly, the reviewer mainly has two concerns. 1: Is RIM in the PSD fraction minimal? 2: Is the band shown in the figures specific? We are sorry that we didn't make it clear in the previous version and happy to give a discussion here.

1: Is Rim in the PSD fraction minimal?

Based on our results, RIM is expressed in PSD, but the amount of RIM in PSD is relative lower compared to presynaptic RIM. However, it's interesting that even its expression in PSD is low, it does play a critical role in regulating postsynaptic functions (supported by Figure 2 and 3 in our manuscript). Indeed, we didn't try to claim the expression level of RIM is high in PSD in this study. What we want to say is that RIM is present in PSD, which is critical for synaptic function. To make this clear, we have revised our text in the current version.

2: Is the band shown in the figures specific?

This is a good and important question. If the band is just a non-specific binding, then RIM may not be present in PSD at all. However, we are confident this is not the case. In our manuscript, we have used two biochemical protocols to get the PSD fraction. One is based on Triton solubility and results from three independent experiments are shown in Figure 1 in this rebuttal letter. The other is based on sucrose gradient and results from 3 independent experiments are shown in Figure 2 in this rebuttal letter. Although the immunoreactivity for RIM1 in the PSD fraction is relatively weak, it can be detected stably (Figure 2). Moreover, the immunoreactivity for RIM1 turns to be stronger when we double the loading sample of the PSD fraction or even increase it five-fold, similar with that of PSD95 (Figures 2b and 2c). In contrast, no immunoreactivity for Synaptophysin, Bassoon, or Munc13 has been detected in the PSD fraction. Since the antibody for RIM has been

validated using IHC in RIM KO mice (Figure S1b), we believe that these experiments strongly support our conclusion for the existence of RIM1 in the PSD fraction.

A new concern is that the authors do not detect bassoon in their PSD preparation. Although bassoon is widely considered to be only present at the presynaptic site, many different studies using the same method for PSD preparations as the one the authors (based on Carlin et al., 1980) clearly find substantial amounts of bassoon in their PSD fractions (e.g., tom Dieck et al., 1998: J Cell Biol 142, 499-509; Walikonis et al., 2000: J Neurosci 20, 4069-4080; Yoshiyuki et al., 2004: J Neurochem 88, 759-768; Lowenthal et al., 2015: J Proteome Res 14, 2528-2538). Although the presence of bassoon is considered to be a contamination, it is highly reproducible and a specific localization at the PSD cannot be ruled out. That the authors do not report a lack of bassoon raises some concern of how their PSD preparation compares to those by many others.

Good point. For various reasons, three of the studies mentioned by the reviewer detect Bassoon in their PSD fractions by using proteomic analysis, which is much more sensitive than biochemical assay. Importantly, one of them also finds RIM1 in their PSD fraction (Yoshiyuki et al., 2004: J Neurochem 88, 759-76). As the reviewer points out, the presence of bassoon in PSD is considered to be a contamination, or it might just be bassoon bound to junctional proteins in the PSD fraction that span the synaptic cleft. Obviously, whether the Bassoon is present in PSD is interesting and warranted for more studies. We are not sure why we didn't observe a band for Bassoon, which could be caused by using different antibodies. Indeed, the only one study detecting Bassoon by biochemical assay mentioned by the reviewer used home-made antibodies (J Cell Biol 142, 499-509, 1998), which is different from ours (Synaptic Systems). Since other presynaptic markers such as synaptophysin have proven the purity of our preparations, we agree that blotting with Bassoon would not help and may even cause confusion. For this reason, we are happy to take the blotting for Bassoon out from the current version.

Figure 1 Subcellular fractionation of the mouse cortex by synaptosome fractionation was probed for presynaptic and postsynaptic marker and RIM1. a, b, and c show the results from three different experiments. H, homogenates; S2, supernatant after first high speed (12000g) centrifugation; P2, pellet after first high speed (12000g) centrifugation.

Figure 2 Subcellular fractionation of the mouse cortex by sucrose gradient centrifugation was probed for presynaptic and postsynaptic marker and RIM1. a, b, and c are results from 3 independent experiments. Total, homogenates; SPM, synaptic plasma membrane fraction; PSD, postsynaptic density fraction. 2F means two-fold, and 5F means five-fold.

2. Figure 1C: the EM labeling for RIM remains to be unconvincing. Fig1 C left is supposedly showing presynaptic labeling for RIM but the label is not very strong and could be due to variability in obtaining contrast. Even worse, the opposing structure is anything but a typical postsynaptic site-it seems a very wide and elongated structure with mitochondria and/or other large organelles –this is certainly not a postsynaptic site of a glutamatergic synapse, which is typically a dendritic spine. In the middle panel the authors identify a dendritic spine with apparently strong immunostaining. However, none of the presynaptic sites in this image are, curiously enough, labeled – this raises serious concerns that the labeling in these images is random and no specific because most if not all presynaptic sites should have RIM.

We appreciate the reviewer’s concern. Indeed, we think that the post-synaptic structure (as indicated by white arrow) in Figure 1C left is a typical dendritic spine with an obvious PSD. In addition, we do not observe the structure in this postsynaptic site as mentioned by the reviewer “... **a very wide and elongated structure with mitochondria and/or other large organelles...**” In contrast, the presynaptic terminal (as indicated by “a”) opposing to this post-synaptic structure is a wide and elongated structure with mitochondria and vesicles. For the middle figures, we just select one only with post-synaptic labeling of RIM1. Actually, we found that RIM1 labeling could be observed either on post- or pre-synaptic profiles or on both of them. We have shown the statistical analysis in our manuscript “*In 212 synapses with nano-gold labeled RIM1, we found that gold particles located on presynaptic or postsynaptic sites in 88.7% (188/212) or 42.4% (90/212) of them, or on both presynaptic and postsynaptic sites in 31.1% (66/212) indicating that*”

RIM1 was distributed both pre- and post-synaptically in area CA1”(p.7, l. 5). Based on our observation, we choose three panels in Figure 1c, with the left one showing only presynaptic labeling of RIM1, the middle one showing only postsynaptic labeling of RIM1, and the right one showing both pre- and post-synaptic labeling of RIM1. We think that the possible reasons why not all presynaptic sites have RIM1 may be due to the limitation of the immunostaining method or the super-thin section of the brain slices. Anyway, we admit that the middle figure is not classical and agree to delete it or move it to the supplemental data if required.

3. Solubilization of NMDARs is not trivial. And centrifugation at 37,000 x g will not bring all membrane fragments into the pellet-one can look this up in any handbook on subcellular fractionation. Now there has been evidence that 1% Triton X-10 solubilizes ~ 85% of NMDARs in the ER-derived microsomal fraction (Blahos and Wenthold, 1996: J Biol Chem 271, 15669-74) and the addition of 0.5% DOC might solubilize most of the remainder, which would minimize the potential for co-IP of RIM simply by association with the same membrane fragments rather than direct binding to the NMDAR complex. However, that all membranes have been solubilized has to be proven by the authors in order to make this claim and that proof requires sedimentation of non-solubilized material by ultracentrifugation. It would be easier for the authors to repeat 1-2 coIP experiments and apply ultracentrifugation at 100,000 x g for 1 h to eliminate this issue. In the absence of either evidence the conclusion that RIM and NMDARs are in the same protein complexes is not well enough supported.

Very good points and thanks for excellent suggestions. Yes, the solubilization of NMDARs is important for our biochemical analysis. Following the reviewer's suggestion, we repeated several key experiments using ultracentrifuge (100,000g for 1 h). As shown in Figure 3 in this rebuttal letter (also Figure S4 in the manuscript), interactions between GluN2B, PSD95, and RIM1 in the tissue extracts strongly support that RIM1 and NMDARs are in the same protein complexes.

Figure 3 Extracts from the adult mouse cortex were immunoprecipitated with antibodies against GluN2B (left), RIM1 (middle), or PSD95 (right), and then blotted with correspondent antibodies. n = 3 independent experiments performed with brain tissue from three adult mice. IP, immunoprecipitation.

Reviewers' comments:

Reviewer #3 (Remarks to the Author):

I am satisfied with the new coIP experiments of NMDRs and RIM1 after a 100,000 x g centrifugation experiment.

I am not satisfied with the current response and data on the RIM localization by sub cellular fractionation and immune-EM. The PSD preparation is not carefully documented and does not match that of many others. A careful analysis that contains all relevant fractions starting with lysate and the P2 and sucrose gradient synaptosomal fractions and finale triton extracted PSD fraction is not provided. Figure 1 shows no enrichment for PSD95 when it should be strongly enriched from lysate and P2 to PSD.

The same is true for the immunoEM analysis. I suggest presentation of galleries of multiple representative images in supplemental figures. Also it would be desirable to use RIM1 KO or Knock down tissue (perhaps after their in vivo injection) to show that the RIM staining is specific in the EM (to be executed in parallel in positive control tissue).

Either cleaner PSD fractionation or more thorough immuno-EM would be sufficient but one can be expected for a journal of the ranks of Nature Communications.

Reviewer #4 (Remarks to the Author):

This manuscript reports very interesting results, potentially of considerable significance. I would recommend a number of revisions, focused mainly on improving the quality of communication.

General:

More paragraph breaks would help readability

"PSD95" should probably be "PSD-95" throughout text.

As typical for speakers whose native language is Chinese, English definite articles cause trouble. "A" and "the" are often mis-inserted or mistakenly omitted.

Statistical tests seem to be properly applied, but it would be better to explain once (or perhaps once in methods and once in legends) that authors use two-sided unpaired t-tests throughout except as noted, and then simply refer to it as "t-test" throughout, except for the occasional paired ones.

In citing numerical results, authors often use too many decimal places, making results less easily understood.

The supplemental material is valuable. In a few cases it perhaps could be included in main

text, but in any case, authors should make more mention of supplemental methods and results in their text (e.g. "...see Supplemental Figure 3").

In legends, authors commonly report "n=3 experiments" while showing single example. If the point is that results were consistent across 3 experiments, that needs to be said; as written, the statement is unintelligible.

Specifics/details:

Title/abstract ok

Introduction:

Top of p 4: "By now, it is found that NMDA trafficking" needs rewriting. Perhaps "Accumulated evidence shows that NMDA..."?

4 lines down, "that involve" should be "involved."

2/3 down, "...our results demonstrated" should be (for example) "...we find"

Results:

p 6, middle, "...GluA1 were located in the PSD fraction," when I look at the blot in 1a, it looks like there's just as much in the "non-PSD" fraction. In Fig 1b, indeed a very modest amount of RIM1 and RIM2 is detected in the PSD, whereas synaptophysin and synapsin are not. However, this result is less clear than it might be, since both synaptophysin and synapsin are associated with presynaptic vesicles, whereas RIMs are presumably linked mainly to the presynaptic membrane. Accordingly the blot would be more persuasive if a presynaptic membrane or cytomatrix marker were shown. This perhaps is the reason for showing a blot that includes Bassoon in the supplemental material. This result should perhaps be added to Figure 1.

p 7, the tissue quality of the immunogold material is uninspiring, but this is to be expected. The data shown in Fig 1 support authors' results, but only moderately. It would be difficult to provide entirely conclusive evidence using this technique without massive additional labor, but the work performed here could be presented to be more convincing. Right now there's very little explanation of methods, other than to cite Chen et al. 2014 and 2015, and Chen 2014 seems to have no relationship to immunogold. It would be important to know how authors define "presynaptic" and "postsynaptic" labeling sites. It would be useful to know from what depth in the embedded tissue blocks the analysis was performed. Ideally it would be nice to see results blindly compared with those from a known pre- or postsynaptic marker, though this might be asking too much for this manuscript.

Next paragraph, should read "... (percentage of RIM1-positive puncta...)". How is a punctum defined? How is colocalization determined? If RIM1 colocalizes with PSD-95, finding some colocalization also with GluN2B is unsurprising, and itself provides almost no evidence for "a relationship of RIM1 with glutamate receptors." Perhaps authors could simply write "...consistent with the idea that RIM1 has a relationship with glutamate receptors...."

p 8, middle, "consistent with our hypothesis" should be moved to the end of the paragraph.

p 10, 2/3 down, should be "...were tested in an object location..."

Reporting control RNAi as "59.69 ± 1.42%" is (at least) one decimal place too many. The same applies throughout this paragraph and in many other parts of text.

Next to last line, should be "...we performed a temporal order...."

p 11 3rd line, should be "...mice in a novel object..."

7th line, the RNAi effect looks like an effect to me, albeit $p > 0.05$. Authors should either give an actual p-value, or qualify their assertion, e.g. "trend but did not reach significance." I have a similar concern for the rest of this paragraph, e.g. (4 lines down) "no significant difference was observed" is true, but obscures what this reader suspects is a real effect; two lines down, "did not alter anxiety-related" exceeds the data (lack of evidence for a significant effect does not prove lack of effect). I'd suggest authors simply qualify this part, concluding e.g. that it had "only weak effects that did not reach a .05 level of significance" or something.

Next paragraph, too many decimal places.

p 12, co-IP for NMDARs but not AMPARs: I am not a biochemist, but my understanding is that AMPARs are typically more freely soluble than NMDARs and less tightly held to PSD complexes. Could this confound the result?

2/3 down, "similar to the NMDAR subunits" borders on wishful thinking. I would delete or reword that.

Put a paragraph before "Coimmunoprecipitation" in the next line, and delete "then."

p. 13, 5th line, should be "observed an interaction." 4th line from bottom, should be "with transfection efficiency."

p 14, delete 'the' several places: 5th line "into HEK 293T cells; 4 lines down "in cultured hippocampal," and 3 lines down "in hippocampal."

I'd put a paragraph break 2 lines down after "Figure 6a)," and replace "firstly" with "first."

p 15 top, too many decimal places.

Bottom paragraph has an unfortunate beginning. I would suggest (for example) "We then examined how overexpression of GFP-tagged... in cultured neurons affected the surface..."

p 16 heading should read "...NMDAR recycling in cultured..."

Next sentence is unreadable. Perhaps "The above results...?"

3 sentences down, should be "using an antibody against the N terminus..."

Bottom sentence should read " Presynaptic RIM1 operates as a Rab3 effector..."

p. 17 2nd line, should read "with the Rab family..."

4th line, should read "Rab4 in rat brain." Next sentence should read "In cortical homogenate...Rab22 coimmunoprecipitated ...subunits GluN1 and...PSD-95, suggesting that..."

3/5ths down the page, should read "...8d). Rab11 also bound to GST-RIM1..."

End of paragraph, should read "different from that mediating the Rab-3-RIM1..."

p. 18 paragraph should begin "SNARE proteins have been implicated in..."
5 sentences down, should be "in cultured."

Contents of the Discussion are generally appropriate, but the wording is poor. I try here, but the final text needs to be corrected/edited by a native English speaker. I will simply write my opinion of the correct wording

p 19 top line "a role for RIM1.'
next line "for RIM1 to mediate."
Next sentence, "Importantly,...determines NMDAR trafficking, but also."
First line next paragraph, "Evidence for ...molecule is based on previous"
Next sentence, "Moreover, a presynaptic....across different synapses (Grabner...)"
Next sentence, "Using electron microscopy... analysis, we confirmed that.... enriched in presynaptic sites."
Next sentence, "However, we found that a."
Next-to-bottom line, "...focused on spinal cord." The suggestion is possible, but this reviewer would speculate that more likely is that the postsynaptic pool is smaller, and was simply neglected in previous work.

p 20
4th line "... (Figure 2). Interestingly..."
2nd line next paragraph, "...as well, and to be ...trafficking, including different..."
next sentence, "...proteins mediate constitutive...NMDAR exocytosis, including" Next sentence, "We here identify RIM1 as another molecule involved...trafficking, supporting the idea that ..recycling shares mechanisms with presynaptic..." We found that ..."
Next sentence, "...is specifically linked to recycled NMDARs, but not to recycled AMPARs, RIM1 may provide a ..."
Bottom line, "We find that.....postsynaptic RIM1, like presynaptic RIM1, binds to....different vesicle-containing..."

p 21 second sentence, "Notably, Rab11 and ...family involved in." Next sentence, "...figure out the exact role of calcium..."
Next paragraph discusses NR2A vs NR2B at some length. In this reviewer's opinion, the developmental switch is exaggerated and NR2B probably predominates even in the adult; in any case, authors find virtually no difference in RIM1 between the two subunits. Given this basically negative result, it seems pointless and distracting to discuss the topic at any length. Accordingly, this part could be substantially shortened or even deleted completely.

p 22, line 9, "ubiquitous, mainly due...NMDARs. However, our results...the possibility of RIM1-dependent...trafficking also at presynaptic sites, and it would be interesting..."
5 lines down, "...Figure 2), confirming that ...RIM1 regulates synaptic function."
4 lines down, "...in the axons of CA1...."

p. 23, 3rd line "...phenotypes. RIM1....NMDAR trafficking has been implicated in neurological diseases such as autism..."

Methods:

p. 24 bottom, "Detailed information..."

p. 25 line 5, "10 days. Other neurons..."

4 lines down, "...mice using protocols adapted from ..."

Image acquisition and analysis: This work should have been performed in a blinded manner, as was reported for other aspects (e.g. electrophysiology and behavior). At this point authors may wish to revise to make this claim, though this reviewer considers it too late to convincingly assert blinded procedures. In future research, authors should be careful to blind their work when appropriate/feasible and to report same in the submitted version of their manuscripts.

p 29, middle, should say "Student's two-sided paired or unpaired t-test was..."

Figure legends:

p 39, Fig 1a, as mentioned in general comments, "n = 3 independent experiments" is of virtually no meaning as written, and serves only to confuse/distract the reader. Authors could (for example) claim in methods and/or in text that all blots were repeated independently in triplicate and all results shown were qualitatively consistent, and/or could say something of that sort 1-2 times in legends.

p 40 bottom, "separated" should be "separate."

Fig 3 p 41, middle, "separated" should be "separate."

p. 42, top 3e, "AMPA-mediated...compared to neighboring..."; 3f, "compared to neighboring..."; 3g, "AMPA-mediated."

p. 43 Fig 4h, "similar time exploring," looks like a possible effect. Perhaps authors could give a p value more informative than "> 0.05"?

Bottom, Fig 5b, "...in transfected HEK293T..."

p. 44 Fig 5g, "into HEK293T..."

Fig 6 heading, "...involved in both constitutive...."

4 lines down, should be "shRNA-resistant plasmid..."

p 45, Fig 6c, was the analysis blinded?

p 46 Fig 8, bottom of page, "Extracts of mouse cortex were ..."

p 47 top "the corresponding antibodies." I see one more (of many "n = 3 independent experiments," a statement without meaning unless further explained.

2 lines down, (d) GST-RIM1 1-399 interacts directly with..."

Supplement:

p 4, middle, "centrifuged again (12,000g)" to yield two fractions. It's not clear which

fraction is which... pellet vs supernatant? or ?

top of p 5 reads as protocol instructions, not as an explanation of what was done. Adjust wording please.

p 7 top, colocalization analysis. How was colocalization assessed?

Supplemental figures:

p 15, S1a seems quite informative. Should it be in the main text?

S1b, "scarified" should be "sacrificed" perhaps? There is no explanation of what we are supposed to conclude from this figure.

S1e, statistical analysis, I don't see any indication of p values?

Reviewer #5 (Remarks to the Author):

Wang et al present in the manuscript entitled "Postsynaptic RIM1 modulates synaptic function by facilitating membrane delivery of recycling NMDARs in hippocampal neurons" a possible new role of the presynaptic protein RIM1. They found an additional postsynaptic localization of RIM1 using different methods such as Western blotting and pre-embedding-immunogold labeling of the hippocampal CA1 region. Further, they propose that RIM1 is required for basal NMDA, not AMPA, -mediated synaptic release and this way contributes to synaptic plasticity and memory formation. They conclude that RIM1 levels in neurons regulate postsynaptic NMDA receptor trafficking.

These findings are striking and shed a complete new light on this typical presynaptic protein. However, also for classical neuronal SNAREs such as SNAP-25 (Lau et al., 2010, Fossati et al., 2015, Salek et al., 2009), Syntaxin-1 (Hussain et al., 2016), VAMP/synaptobrevin (Hussein and Davanger, 2015) postsynaptic localizations were found and roles in postsynaptic regulation of glutamate receptors suggested. Still, the presented results in this manuscript would certainly start a debate in the field on RIM1 function.

The manuscript is carefully prepared, methods are well described and the authors present a row of experiments investigating the role of RIM1 at the postsynapse in detail. I will concentrate my review specifically on the ultrastructural investigations since here some questions are still open.

The immunogold labeling method is well established and the structural preservation is excellent, because one always has to take into account that the structural preservation is often not as good as for pure morphological studies due to harsh treatments for permeabilization and the lack or reduction of glutaraldehyde as a fixative. Very important is the information the authors already provided, the percentage of how many synapses were labeled either pre- or postsynaptically or both. Therefore, I find the selection of images for figure 1 depicting each possible situation valid and representative. Please state this clearly in the main text (with reference to the figure) and also in the

legend. Further, please explain in the legend, what the white arrows mean (I see that it is the PSD, but please mention).

-Regarding the images I have no problems with the selected synaptic profiles left and in the middle of figure 1c, in all cases a PSD and an opposing presynaptic bouton are visible. However, the right panel is not ideal, a postsynaptic labeling is shown (white arrows pointing to the PSD), but the opposing presynapse is not labeled, rather the one more left, which might not project onto the spine, at least it is a bit questionable to me. This raises the question, how the selection/quantification was done: Was a synapse treated as a synapse having a clear pre- and postsynapse and it was checked whether either one or both were labeled? Or did the authors quantify for example all visible profiles with synaptic vesicles separately?

Related to that comment:

-I assume, more than one animal was prepared (I cannot find the number of animals for this analysis, please clearly mention), therefore it should be described how the sampling was done. How were the synapses "selected"? Was the selection random, or was each possible synapse on one section imaged? Were different tissue sections used to reach different tissue depth? Was the number of synapses more or less equal for each animal?

-Was any other selection done, for example only synapses were selected that showed a clear PSD and synaptic cleft together with an opposing presynapse (and vesicles) or anything like that (see comment above)?

-How was the background determined? I see in the right panel and also in the middle a staining of the dendrite, can the authors please comment on that and how they interpret this staining? Was this also counted as a postsynaptic labeling, or only when a labeling was found at the PSD?

-Was the antibody tried on the RIM1/2 double KOs? Could there be any cross reaction with RIM2?

Since a convincing postsynaptic ultrastructural localization would clearly support their findings, the authors should comment on the raised concerns.

Reviewer #3 (Remarks to the Author):

I am satisfied with the new coIP experiments of NMDRs and RIM1 after a 100,000 x g centrifugation experiment.

We appreciate that the reviewer thinks we have satisfactorily addressed this concern.

I am not satisfied with the current response and data on the RIM localization by sub cellular fractionation and immune-EM. The PSD preparation is not carefully documented and does not match that of many others. A careful analysis that contains all relevant fractions starting with lysate and the P2 and sucrose gradient synaptosomal fractions and finale triton extracted PSD fraction is not provided. Figure 1 shows no enrichment for PSD95 when it should be strongly enriched from lysate and P2 to PSD.

Response

1. We are sorry for not clearly describing our protocol in the previous manuscript. In the revised manuscript, we reword in detail the biochemical protocols we used to obtain the PSD fraction. Please see p.5-6 in the supplementary materials and methods.
2. In our last rebuttal letter, we provided results from three independent experiments for both of the PSD preparation protocols used, and the results are consistent

throughout. We also provide data using different presynaptic and postsynaptic markers to confirm our protocol. Therefore, we believe that our results are sufficiently convincing.

3. We appreciate the reviewer's concern. That PSD-95 shows no obvious enrichment from lysate and P2 to PSD in Figure 1a is mainly because we did not measure the concentrations in these fractions and just loaded similar volumes of protein extracts. When specific amounts of protein extracts from different fractions (20 μ g of homogenate, 10 μ g of each of S2, P2, the non-PSD fraction, and the PSD fraction) were loaded, enrichment of PSD-95 in the PSD fraction was clearly evident as shown in **Figure 1** in this letter.

Figure 1 Subcellular fractionation of mouse cortex by synaptosome fractionation was probed for the postsynaptic marker PSD-95 and RIM1. H, homogenate; S2, supernatant after first high-speed (12,000g) centrifugation; P2, pellet after first high-speed (12,000g) centrifugation.

The same is true for the immunoEM analysis. I suggest presentation of galleries

of multiple representative images in supplemental figures. Also it would be desirable to use RIM1 KO or Knock down tissue (perhaps after their in vivo injection) to show that the RIM staining is specific in the EM (to be executed in parallel in positive control tissue).

Response

1. We appreciate the reviewer's suggestion. We show galleries of representative images as a supplemental figure in the revised manuscript (Figure S1). We also show these results as Figure 2 in this letter.
2. We appreciate the reviewer's concern. We have tested the specificity of RIM1 staining in the IHC by using the hippocampus of RIM1^{floxed} mice injected with AAV-hSyn-Cre-GFP in our previous manuscript. As shown in Figure S2a in the revised manuscript, GFP-positive cells showed much lower RIM1 fluorescence intensity than GFP-negative cells. We believe this result also show that RIM1 staining is specific.

Figure 2. Gallery of electron micrographs of RIM1 immunogold staining.

a-c, Nano-gold-labeled RIM1 distributed in presynaptic axon terminals. d-f, Nano-gold-labeled RIM1 distributed in postsynaptic dendrites. g-i, Nano-gold-labeled RIM1 distributed in both presynaptic and postsynaptic profiles. Blue arrows, synapses with nano-gold in presynaptic profiles; red arrows, synapses with nano-gold in postsynaptic profiles; green arrows, synapses with nano-gold in both pre- and postsynaptic profiles. Scale bar, 500 nm in a-e and g-h; 250 nm in f and i.

Either cleaner PSD fractionation or more thorough immuno-EM would be sufficient but one can be expected for a journal of the ranks of Nature Communications.

Response

We do hope that the reviewer is satisfied with our new data and explanation.

Reviewer #4 (Remarks to the Author):

This manuscript reports very interesting results, potentially of considerable significance. I would recommend a number of revisions, focused mainly on improving the quality of communication.

We appreciate the positive comments, and we also appreciate the reviewer's careful work to improve our writing.

General:

More paragraph breaks would help readability

Response

Thanks for the suggestion. We have asked a native English speaker to re-polish our revised manuscript.

"PSD95" should probably be "PSD-95" throughout text.

Response

Thanks for the suggestion. We have corrected this in the revised manuscript.

As typical for speakers whose native language is Chinese, English definite articles cause trouble. "A" and "the" are often mis-inserted or mistakenly omitted.

Response

We apologize for misusing “A” and “the”. We have asked a native English speaker to read through our manuscript and correct this.

Statistical tests seem to be properly applied, but it would be better to explain once (or perhaps once in methods and once in legends) that authors use two-sided unpaired t-tests throughout except as noted, and then simply refer to it as "t-test" throughout, except for the occasional paired ones.

Response

Thanks for the suggestion. We have explained twice (once in the Methods and once in the figure legend, p35, line 21, and p55, the last line in the main text) that we use

two-sided unpaired t-tests throughout except as noted, and then refer to it as “t-test” throughout.

In citing numerical results, authors often use too many decimal places, making results less easily understood.

Response

Thanks for the suggestion. We have rewritten our numerical results and tried to use no decimal places or only one decimal place throughout to make the results easily understood.

The supplemental material is valuable. In a few cases it perhaps could be included in main text, but in any case, authors should make more mention of supplemental methods and results in their text (e.g. "...see Supplemental Figure 3").

Response

We appreciate the reviewer's considerate suggestion. We have modified the manuscript correspondingly as suggested and highlighted some of the supplemental data as needed.

In legends, authors commonly report "n=3 experiments" while showing single

example. If the point is that results were consistent across 3 experiments, that needs to be said; as written, the statement is unintelligible.

Response

Thanks for the suggestion. We have deleted the relevant statements in the revised manuscript.

Specifics/details:

Title/abstract ok

Introduction:

Top of p 4: "By now, it is found that NMDA trafficking" needs rewriting.

Perhaps "Accumulated evidence shows that NMDA..."?

4 lines down, "that involve" should be "involved."

2/3 down, "...our results demonstrated" should be (for example) ...we find"

Response

Thanks for the suggestion. We have corrected these points correspondingly.

Results:

p 6, middle, "...GluA1 were located in the PSD fraction," when I look at the blot

in 1a, it looks like there's just as much in the "non-PSD" fraction.

Response

We agree that GluA1, unlike NMDAR subunits, is also distributed in the non-PSD fraction. Therefore, we rewrote the sentence as “...*The AMPA receptor subunit GluA1 was located in both the non-PSD and PSD fractions...*” in the revised manuscript (p7, line 12 in the main text).

In Fig 1b, indeed a very modest amount of RIM1 and RIM2 is detected in the PSD, whereas synaptophysin and synapsin are not. However, this result is less clear than it might be, since both synaptophysin and synapsin are associated with presynaptic vesicles, whereas RIMs are presumably linked mainly to the presynaptic membrane. Accordingly the blot would be more persuasive if a presynaptic membrane or cytomatrix marker were shown. This perhaps is the reason for showing a blot that includes Bassoon in the supplemental material. This result should perhaps be added to Figure 1.

Response

We appreciate the reviewer’s thoughtful suggestion, and we agree that an appropriate marker may help to make our data more persuasive. RIMs are adaptors linking presynaptic vesicles with other presynaptic membrane proteins, such as VGCCs. Based on the function and synaptic localization of presynaptic proteins, we prefer to

use Munc13 and Bassoon as markers. Combined with the reviewer's suggestion in the following, we added a blot that includes Munc13 to the previous Supplementary Figure 1a, and then moved it to Figure 1 as Figure 1c.

We also show the panel as Figure 3 in this letter.

Figure 3 Subcellular fractionation of mouse cortex by sucrose gradient centrifugation was probed for the presynaptic markers synaptophysin, Munc13, and Bassoon, the postsynaptic marker PSD-95, and RIM1. Total, homogenates; SPM, synaptic plasma membrane fraction; PSD, postsynaptic density fraction, 1F and 2F represent the normal and the 2-fold quantity of the PSD fraction sample, respectively.

p 7, the tissue quality of the immunogold material is uninspiring, but this is to be expected. The data shown in Fig 1 support authors' results, but only moderately. It would be difficult to provide entirely conclusive evidence using this technique without massive additional labor, but the work performed here could be presented to be more convincing. Right now there's very little explanation of methods, other than to cite Chen et al. 2014 and 2015, and Chen 2014 seems to

have no relationship to immunogold. It would be important to know how authors define "presynaptic" and "postsynaptic" labeling sites.

Response

We appreciate the reviewer's concern. In the present version, we described the main steps of the pre-embedding EM procedure in the Methods section (p29-31 in the main text). The cited refs were also changed to *Chen et al., Mol Brain, 2014* and *Chen et al., Mol Neurobiol, 2016*, in which nano-gold immunostaining was used. We only selected typical synapses for sample pictures and statistics, in which presynaptic structures (with vesicles), postsynaptic structures (with clear PSDs), and synaptic clefts were all clearly visible.

We also present a gallery of EM images showing the RIM1 labeling in pre- and postsynaptic profiles for further confirmation (see Figure 2 in this letter or supplementary Figure 1 in the revised manuscript)

It would be useful to know from what depth in the embedded tissue blocks the analysis was performed. Ideally it would be nice to see results blindly compared with those from a known pre- or postsynaptic marker, though this might be asking too much for this manuscript.

Response

Thanks for the reviewer's concern.

1. We agree that the depth in the embedded tissue is important since only the superficial layers of the tissue blocks are well stained in the EM process. We thus only cut ~20 ultrathin (70 nm) sections beginning from the surface of the tissue block and selected 10-15 of those with good RIM1 immunoreactivity for imaging. Thus, we think all the selected structures are similar in terms of tissue depth and RIM1 immunoreactivity.
2. We appreciate the reviewer's thoughtful suggestion. It would be much better to compare our results with those from a known pre- or postsynaptic marker. Here, the location of RIM1 is determined by synaptic structure, which is also a reliable approach. We will take this into consideration when we design experiments in the future.

Next paragraph, should read "...(percentage of RIM1-positive puncta...". How is a punctum defined? How is colocalization determined? If RIM1 colocalizes with PSD-95, finding some colocalization also with GluN2B is unsurprising, and itself provides almost no evidence for "a relationship of RIM1 with glutamate receptors." Perhaps authors could simply write "..consistent with the idea that RIM1 has a relationship with glutamate receptors...."

Response

Thanks for the reviewer's concern.

1. Synaptic puncta were determined by a threshold set at a factor of 2-4 times the

average dendritic gray value. Puncta were counted by MetaMorph 7.5 software (Universal Imaging) and then double-checked manually. We added these descriptions to the revised manuscript (p10, line 17 in the supplemental materials and methods).

2. For colocalization analysis, images were separately thresholded for each channel followed by a minimal size cut-off; colocalization of two fluorescent signals was determined using the “colocalization” module in MetaMorph. This module provides the area measurements of the region overlap between signals in red and green channels of image projections. We added these descriptions to the revised manuscript (p10, line 20 in the supplemental materials and methods)
3. We agree with the reviewer and have modified the sentence as suggested.

p 8, middle, "consistent with our hypothesis" should be moved to the end of the paragraph.

p 10, 2/3 down, should be "...were tested in an object location..."

Reporting control RNAi as "59.69 ± 1.42% is (at least) one decimal place too many. The same applies throughout this paragraph and in many other parts of text.

Next to last line, should be "...we performed a temporal order..."

Response

Thanks for the valuable suggestion. We have corrected these points correspondingly.

p 11 3rd line, should be "...mice in a novel object..."

7th line, the RNAi effect looks like an effect to me, albeit $p > 0.05$. Authors should either give an actual p-value, or qualify their assertion, e.g. "trend but did not reach significance." I have a similar concern for the rest of this paragraph, e.g. (4 lines down) "no significant difference was observed" is true, but obscures what this reader suspects is a real effect; two lines down, "did not alter anxiety-related" exceeds the data (lack of evidence for a significant effect does not prove lack of effect). I'd suggest authors simply qualify this part, concluding e.g. that it had "only weak effects that did not reach a .05 level of significance" or something.

Next paragraph, too many decimal places.

Response

We appreciate these valuable suggestions. We have modified our manuscript as suggested and simplified the numerical results. In addition, we have added the p values for Figure 4h and 4j (p13, line 11 and 16 in the main text).

p 12, co-IP for NMDARs but not AMPARs: I am not a biochemist, but my understanding is that AMPARs are typically more freely soluble than NMDARs and less tightly held to PSD complexes. Could this confound the result?

Response

Good point. We agree that AMPARs are more freely soluble and dynamic, while NMDARs are more static and tightly associated with PSD complexes. In this study, we performed co-IP experiments on NMDARs and RIM1 by ultracentrifugation (**100,000 × g**) to sediment the non-solubilized material (Figure S5 in the manuscript). Therefore, it is impossible that differences in solubility confounded our results. There is still another possibility that the interaction between AMPAR and RIM1 is too dynamic to be detected by coIP. However, combined with our electrophysiological recordings showing that RIM1 KD has no effect on AMPAR function, this possible dynamic interaction is functionally unimportant.

2/3 down, "similar to the NMDAR subunits" borders on wishful thinking. I would delete or reword that.

Put a paragraph before "Coimmunoprecipitation" in the next line, and delete "then."

Response

Thanks for the suggestion. We deleted “*similar to the NMDAR subunits*”, and rewrote the following sentence as “*To investigate the interaction of RIM1 with different NMDAR subunits, we cotransfected HA-tagged RIM1 (HA-RIM1) with GFP-tagged NMDAR subunits (GFP-GluN1, GFP-GluN2A, or GFP-GluN2B) into HEK293T cells.*”

(p15, line 3 in the main text)

p. 13, 5th line, should be "observed an interaction." 4th line from bottom, should be "with transfection efficiency."

Response

Thanks for the suggestion; we have corrected these points according to the reviewer's suggestions.

p 14, delete 'the' several places: 5th line "into HEK 293T cells; 4 lines down "in cultured hippocampal," and 3 lines down "in hippocampal."

I'd put a paragraph break 2 lines down after "Figure 6a)," and replace "firstly" with "first."

Response

Thanks for the suggestion; we have corrected these points.

p 15 top, too many decimal places.

Bottom paragraph has an unfortunate beginning. I would suggest (for example) "We then examined how overexpression of GFP-tagged... in cultured neurons affected the surface..."

Response

Thanks for the suggestion. We have corrected these points.

p 16 heading should read "...NMAR recycling in cultured...."

Next sentence is unreadable. Perhaps "The above results..."?"

3 sentences down, should be "using an antibody against the N terminus..."

Bottom sentence should read " Presynaptic RIM1 operates as a Rab3 effector..."

Response

Thanks for these valuable suggestions. We have corrected these points.

p. 17 2nd line, should read "with the Rab family..."

4th line, should read "Rab4 in rat brain." Next sentence should read "In cortical homogenate...Rab22 coimmunoprecipitated ...subunits GluN1 and...PSD-95, suggesting that..."

3/5ths down the page, should read "...8d). Rab11 also bound to GST-RIM1..."

End of paragraph, should read "different from that mediating the Rab-3-RIM1..."

Response

Thanks for these valuable suggestions. We have corrected these points.

p. 18 paragraph should begin "SNARE proteins have been implicated in..."

5 sentences down, should be "in cultured."

Response

Thanks for these suggestions. We have corrected these points.

Contents of the Discussion are generally appropriate, but the wording is poor. I try here, but the final text needs to be corrected/edited by a native English speaker. I will simply write my opinion of the correct wording

Response

We really appreciate the reviewer's valuable suggestion and we have asked a native English speaker to help us re-polish it.

p 19 top line "a role for RIM1.'

next line "for RIM1 to mediate."

Next sentence, "Importantly,...determines NMDAR trafficking, but also."

First line next paragraph, "Evidence for ...molecule is based on previous"

Next sentence, "Moreover, a presynaptic....across different synapses (Grabner...)"

Next sentence, "Using electron microscopy... analysis, we confirmed that... enriched in presynaptic sites."

Next sentence, "However, we found that a."

Response

Thanks to the valuable suggestions. We have corrected these points in the revised manuscript.

Next-to-bottom line, "...focused on spinal cord." The suggestion is possible, but this reviewer would speculate that more likely is that the postsynaptic pool is smaller, and was simply neglected in previous work.

Response

Good point. We have added a sentence to this paragraph as "*It is also possible that the postsynaptic pool is smaller and was neglected in previous work.*" (p24, line 2 in the main text)

p 20

4th line "...(Figure 2). Interestingly..."

2nd line next paragraph, "..as well, and to be ...trafficking, including different..."

next sentence, "..proteins mediate constitutive...NMDAR exocytosis,

including" Next sentence, "We here identify RIM1 as another molecule

involved...trafficking, supporting the idea that ..recycling shares mechanisms

with presynaptic..." We found that ..."

Next sentence, "...is specifically linked to recycled NMDARs, but not to recycled

AMPARs, RIM1 may provide a ..."

Bottom line, "We find that.....postsynaptic RIM1, like presynaptic RIM1, binds to....different vesicle-containing..."

Response

Thanks for these valuable suggestions. We have corrected these points in the revised manuscript.

p 21 second sentence, "Notably, Rab11 and ...family involved in." Next sentence, "...figure out the exact role of calcium..."

Response

Thanks for the suggestions. We have corrected these points.

Next paragraph discusses NR2A vs NR2B at some length. In this reviewer's opinion, the developmental switch is exaggerated and NR2B probably predominates even in the adult; in any case, authors find virtually no difference in RIM1 between the two subunits. Given this basically negative result, it seems pointless and distracting to discuss the topic at any length. Accordingly, this part could be substantially shortened or even deleted completely.

Response

Thanks for the suggestion. In the revised manuscript, we have shortened this paragraph and deleted the sentence about the developmental switch of NMDARs (p25-26 in the main text).

p 22, line 9, "ubiquitous, mainly due...NMDARs. However, our results...the possibility of RIM1-dependent...trafficking also at presynaptic sites, and it would be interesting..."

5 lines down, "...Figure 2), confirming that ...RIM1 regulates synaptic function.

4 lines down, "...in the axons of CA1..."

p. 23, 3rd line "...phenotypes. RIM1...NMDAR trafficking has been implicated in neurological diseases such as autism..."

Response

Thanks for the suggestions. We have corrected them in the revised manuscript.

Methods:

p. 24 bottom, "Detailed information..."

p. 25 line 5, "10 days. Other neurons..."

4 lines down, "...mice using protocols adapted from ..."

Response

Thanks for the suggestions. We have corrected them in the revised manuscript.

Image acquisition and analysis: This work should have been performed in a blinded manner, as was reported for other aspects (e.g. electrophysiology and behavior). At this point authors may wish to revise to make this claim, though this reviewer considers it too late to convincingly assert blinded procedures. In future research, authors should be careful to blind their work when appropriate/feasible and to report same in the submitted version of their manuscripts.

Response

We greatly appreciate the reviewer's valuable and thoughtful suggestion. We did not perform the imaging experiments in a blinded manner, although we usually asked another student to repeat the imaging experiments.

We will keep this in mind and will design our imaging experiment in a blinded way in future. Thanks again for your consideration.

p 29, middle, should say "Student's two-sided paired or unpaired t-test was..."

Response

Thanks for the suggestions. We have corrected them in the revised manuscript.

Figure legends:

p 39, Fig 1a, as mentioned in general comments, "n = 3 independent experiments" is of virtually no meaning as written, and serves only to confuse/distract the reader. Authors could (for example) claim in methods and/or in text that all blots were repeated independently in triplicate and all results shown were qualitatively consistent, and/or could say something of that sort 1-2 times in legends.

Response

Thanks for the suggestion. We added "*All blots were repeated independently in triplicate and all the results shown were qualitatively consistent.*" to the supplemental material and methods (p7, line 7).

p 40 bottom, "separated" should be "separate."

Fig 3 p 41, middle, "separated" should be "separate."

p. 42, top 3e, "AMPA-mediated...compared to neighboring..."; 3f, "compared to neighboring..."; 3g, "AMPA-mediated."

Response

Thanks for the suggestions. We have corrected them in the revised manuscript.

p. 43 Fig 4h, "similar time exploring," looks like a possible effect. Perhaps authors could give a p value more informative than "> 0.05"?

Response

Thanks for the suggestions. We have given the p value ($p = 0.34$) both in the main text (p13, line 11) and in the figure legend (p58, line 18) in the revised manuscript.

Bottom, Fig 5b, "...in transfected HEK293T..."

p. 44 Fig 5g, "into HEK293T..."

Fig 6 heading, "...involved in both constitutive...."

4 lines down, should be "shRNA-resistant plasmid..."

Response

Thanks for the suggestions. We have corrected them in the revised manuscript.

p 45, Fig 6c, was the analysis blinded?

Response

No, the analysis was not blinded.

p 46 Fig 8, bottom of page, "Extracts of mouse cortex were ..."

p 47 top "the corresponding antibodies." I see one more (of many "n = 3 independent experiments," a statement without meaning unless further

explained.

2 lines down, (d) GST-RIM1 1-399 interacts directly with..."

Response

Thanks for the suggestions. We have corrected them in the revised manuscript.

Supplement:

p 4, middle, "centrifuged again (12,000g)" to yield two fractions. It's not clear which fraction is which... pellet vs supernatant? or ?

Response

We are sorry for not describing this clearly. In the revised manuscript, we reword the protocol as *"To further digest the synaptosomes and yield an insoluble "PSD-enriched" membrane fraction and a "non-PSD-enriched" membrane fraction, we resuspended the P2 pellet in 4 mM HEPES buffer (4 mM HEPES and 1 mM EDTA, pH 7.4) and centrifuged again (12,000 g, 20 min, 4°C). Resuspension and centrifugation were repeated. The resulting pellet was then resuspended with buffer A (20 mM HEPES, 100 mM NaCl, 0.5% Triton X-100, pH 7.2) and rotated slowly (15 min, 4°C), followed by centrifugation (12,000 g, 20 min, 4°C). The supernatant containing Triton X-100-soluble non-PSD membranes was retained. The pellet was resuspended in buffer B (20 mM HEPES, 0.15 mM NaCl, 1% TritonX-100, 1% deoxycholic acid, 1% sodium dodecyl sulfate, and 1 mM dithiothreitol, pH 7.5),*

followed by gentle rotation (1 h, 4°C) and centrifugation (10,000 g, 15 min, 4°C). The pellet was discarded, and the supernatant (Triton X-100-insoluble PSD fraction) was retained.” (p5, line5 in the supplemental materials).

top of p 5 reads as protocol instructions, not as an explanation of what was done.

Adjust wording please.

Response

Thanks for the suggestion. We have modified this in the revised manuscript “*The pellet (crude synaptosomal fraction, P2) was resuspended in 0.32 M HEPES-buffered sucrose and rotated for 30 min at 4°C to ensure complete lysis. After centrifugation at 10,000 g for another 15 min, the pellet fraction (P2’) was resuspended with 4-fold ddH₂O and homogenized with 3 strokes of a glass-Teflon homogenizer. Then, the solution was adjusted to 4 mM HEPES and rotated at 4°C for 30 min. After centrifugation at 25,000 g for 20 min (4°C), the pellet (synaptosomal membrane fraction) was resuspended with 0.32 M HEPES-buffered sucrose and laid on a discontinuous sucrose gradient containing 0.8 to 1.0 to 1.2 M sucrose. After another centrifugation at 150,000 g for 2 h (4°C) using an SW41Ti rotor (Beckman), the synaptic plasma membrane was collected from a cloudy band between 1.0 M and 1.2 M sucrose. The suspension was diluted to 0.32 M and further centrifuged at 200,000 g for 30 min at 4°C using an MLA-150 rotor (Beckman).*” (p6, the first line in the supplemental materials).

p 7 top, colocalization analysis. How was colocalization assessed?

Response

We are sorry for not describing it clearly. For colocalization analysis, images were separately thresholded for each channel followed by a minimal size cut-off; colocalization of two fluorescent signals was determined using the “colocalization” module in MetaMorph. This module provides the area measurements of the region overlap between signals in red and green channels of image projections.

We have added this information to the supplemental materials (p10-11)

Supplemental figures:

p 15, S1a seems quite informative. Should it be in the main text?

Response

We appreciate the reviewer’s suggestion. In the revised manuscript, we have moved it to Figure 1 as Figure 1c

S1b, "scarified" should be "sacrificed" perhaps? There is no explanation of what we are supposed to conclude from this figure.

Response

Thanks for the suggestion; we have corrected it.

We are sorry for not making this clear. This experiment was designed to test the specificity of RIM1 staining. We injected AAV-hSyn-Cre-GFP into the hippocampus of RIM1^{floxed} mice. Three weeks after injection, the hippocampus was sectioned and stained with antibody against RIM1. As shown in this panel, the fluorescent intensity of RIM1 in GFP-positive cells was much lower than that in the GFP-negative cells, indicating that the RIM1 staining is specific in our system.

S1e, statistical analysis, I don't see any indication of p values?

Response

We are sorry for the mistake. It should be “quantitative analysis”

Reviewer #5 (Remarks to the Author):

Wang et al present in the manuscript entitled “Postsynaptic RIM1 modulates synaptic function by facilitating membrane delivery of recycling NMDARs in hippocampal neurons” a possible new role of the presynaptic protein RIM1. They found an additional postsynaptic localization of RIM1 using different methods such as Western blotting and pre-embedding-immunogold labeling of the hippocampal CA1 region. Further, they propose that RIM1 is required for basal NMDA, not AMPA, -mediated synaptic release and this way contributes to

synaptic plasticity and memory formation. They conclude that RIM1 levels in neurons regulate postsynaptic NMDA receptor trafficking.

These findings are striking and shed a complete new light on this typical presynaptic protein. However, also for classical neuronal SNAREs such as SNAP-25 (Lau et al., 2010, Fossati et al., 2015, Salek et al., 2009), Syntaxin-1 (Hussain et al., 2016), VAMP/synaptobrevin (Hussein and Davanger, 2015) postsynaptic localizations were found and roles in postsynaptic regulation of glutamate receptors suggested. Still, the presented results in this manuscript would certainly start a debate in the field on RIM1 function.

We thank the reviewer for the thoughtful and positive comments. Indeed, we also anticipate that our work will be of great interest to those concerned with synaptic function, organization, and plasticity.

The manuscript is carefully prepared, methods are well described and the authors present a row of experiments investigating the role of RIM1 at the postsynapse in detail. I will concentrate my review specifically on the ultrastructural investigations since here some questions are still open.

We thank the reviewer for these positive comments.

The immunogold labeling method is well established and the structural preservation is excellent, because one always has to take into account that the structural preservation is often not as good as for pure morphological studies due to harsh treatments for permeabilization and the lack or reduction of glutaraldehyde as a fixative.

We completely agree with the reviewer.

Very important is the information the authors already provided, the percentage of how many synapses were labeled either pre- or postsynaptically or both. Therefore, I find the selection of images for figure 1 depicting each possible situation valid and representative. Please state this clearly in the main text (with reference to the figure) and also in the legend. Further, please explain in the legend, what the white arrows mean (I see that it is the PSD, but please mention).

Response

We appreciate the reviewer's suggestion. We have stated this in the main text (p8, line 6) and also in the legend (p54, the last line).

-Regarding the images I have no problems with the selected synaptic profiles left and in the middle of figure 1c, in all cases a PSD and an opposing presynaptic

bouton are visible. However, the right panel is not ideal, a postsynaptic labeling is shown (white arrows pointing to the PSD), but the opposing presynapse is not labeled, rather the one more left, which might not project onto the spine, at least it is a bit questionable to me. This raises the question, how the selection/quantification was done: Was a synapse treated as a synapse having a clear pre- and postsynapse and it was checked whether either one or both were labeled? Or did the authors quantify for example all visible profiles with synaptic vesicles separately?

Response

For the right panel of Fig. 1c, we are very sorry for the confusion because of our carelessness. Actually, both the left and right axon terminals made synapses with the dendrite. We just failed to indicate the left synapse with an arrow (you can see the clear PSD partially covered by a nanogold product). We have marked the left synapse in the new version (Figure 1d in the revised manuscript).

The synapses we selected for sample pictures and statistics were only those with clear presynaptic axon terminals (with vesicles), postsynaptic structures (with PSDs) and synaptic clefts.

Related to that comment:

-I assume, more than one animal was prepared (I cannot find the number of

animals for this analysis, please clearly mention), therefore it should be described how the sampling was done. How were the synapses “selected”? Was the selection random, or was each possible synapse on one section imaged? Were different tissue sections used to reach different tissue depth? Was the number of synapses more or less equal for each animal?

Response

We appreciate these concerns. We have checked the RIM1 immunoreactivity in 2 mice from the previous version of manuscript and added 2 more to the present version (4 mice in total). Around 20 super-thin (70 nm) sections were cut beginning from the surface of each hippocampal sample block and finally 10-15 sections with good RIM1 immunoreactivity in each mouse were selected for imaging. We randomly selected ~10 typical synapses with RIM1 immunoreactivity in each section and selected a total of ~100 synapses for each mouse.

We have described these in the Methods section in the new version of the manuscript (p29-31 in the main text).

-Was any other selection done, for example only synapses were selected that showed a clear PSD and synaptic cleft together with an opposing presynapse (and vesicles) or anything like that (see comment above)?

Response

Only synapses with clear PSDs, synaptic clefts, and opposing presynaptic structures (with vesicles) were selected for sample images and statistics.

-How was the background determined? I see in the right panel and also in the middle a staining of the dendrite, can the authors please comment on that and how they interpret this staining? Was this also counted as a postsynaptic labeling, or only when a labeling was found at the PSD?

Response

We understand the reviewer's concern about the possible mixed background and nanogold immunoreactivity. From the image shown, it may be difficult to distinguish them. However, the background and immunoreactive products can be clearly distinguished by changing the brightness/contrast of the images. As shown in the following images (Figure 4 in this letter), if we greatly increase the brightness of Fig. 1c, we can easily distinguish them: the background staining becomes very weak but the nanogold immunoreactivity is much less affected.

In some cases, the trouble originated, at least partially, from our uncertainty about showing the sample images with the best brightness/contrast. We have slightly increased the brightness of the sample images for better illustration in the new Fig. 1c. Normally we do not change the original brightness/contrast.

Figure 4 Images of Figure 1d (in the main text) with greater brightness

-Was the antibody tried on the RIM1/2 double KOs? Could there be any cross reaction with RIM2?

Response

We appreciate the reviewer's concern. We haven't tried this antibody on RIM1/2 double KOs. But we have tested it on RIM2 KO mice and on transfection system. As shown in Figure 5A in this letter, the immunoreactivity of RIM1 showed no evident decrease in the brain homogenate from RIM2 KO mice compared with that from wild-type mice. Moreover, no RIM1 signal was detected when HEK293 cells were transfected with HA-RIM2 (Figure 5B in this letter). Thus, it indicates that this antibody has no cross-reaction with RIM2.

Figure 5 Specificity of RIM1 antibodies from SYSY Company. Brain homogenate from wild-type (WT) mice and RIM2 KO mice were used to test the specificity of antibody against RIM1. B HEK293 cells were transfected with vector, HA-tagged RIM1 or HA-tagged RIM2 plasmids. The homogenate of cells were used to test the specificity of antibodies against RIM1

Since a convincing postsynaptic ultrastructural localization would clearly support their findings, the authors should comment on the raised concerns.

Response

We appreciate the reviewer's suggestion. In the revised manuscript, we have added explanations of the concerns raised in the main text, figure legends, and Methods section.

REVIEWERS' COMMENTS:

Reviewer #4 (Remarks to the Author):

This manuscript provides extensive new data suggesting a significant functional role for postsynaptic RIM1 in regulating NMDAR signaling in hippocampus. Results overall are self-consistent and reasonably convincing. My generally substantial enthusiasm for this work is somewhat reduced by several remaining concerns, some which can be easily addressed. The text should be improved in several places, especially by providing a shorter and more focused Discussion. In several places I think authors oversell their work: They provide good evidence for their core findings; in this reader's opinion making claims beyond their data only weakens their argument. I provide specific details and suggestions below. Authors were careful to blind parts of the study (electrophysiology and behavior); it's unfortunate that they apparently failed to blind other parts, but I don't consider that a fatal flaw.

SPECIFICS/DETAILS:

I proceed in sequence through text, for my convenience and that of authors.

ABSTRACT

#30, replace "whereas" with "but."

#37 "determine" is inappropriately used here and several other places through the text. Perhaps authors don't quite understand the connotations of that word, but the unfortunate consequence is to considerably exaggerate the findings. Here, one could write (for example) "...RIM1 levels...influence both constitutive and ..."

INTRODUCTION

#48 delete "the regulation of."

#51, should read "...that lead to rapid insertion of AMPARs into..."

#59, insert "may" between "or" and "be sorted."

#81/82, should read "...proteins, including SNAP25, ..."

#97, put a comma after "N-terminus."

#98-99, the last sentence is too strong. Perhaps authors are concerned that Nature Methods will review their manuscript; at this point they should instead try to provide a realistic and balanced view of their findings for the benefit of potential readers. I would say (for example) "...our results identify a substantial role for postsynaptic...and suggest that this mechanism is important for ..."

RESULTS

#117, replace "that couples" with "which couples."

#126, I am no biochemist, but it seems that the RIM1 signal (e.g. 1b) is weak. Per se this does not detract from the main result, but if authors want to argue that a rather modest amount of postsynaptic RIM1 plays an important functional role (which is what I take to be their message) it might be best to acknowledge this explicitly from the beginning, and say (for example) "but also in the PSD fraction, though at lower levels."

#133, should read "...326 (79.5%) of presynaptic... and on 189 (46.1%) of postsynaptic..."

The material shown in Supplemental figure 1 is ok, but is reproduced too dark; authors should adjust the tonal balance so that the silver-enhanced gold is immediately obvious.

#146, I'm a little puzzled by the punctate colocalization work. Assessments of this type are technically difficult and should have been performed in a blinded manner, but I'm not sure what the goal was in any case. Is this really a satisfactory way to determine whether RIM1 is at the active zone or at the PSD? Perhaps authors have a different model in mind, in which RIM1 is in a cytoplasmic compartment? Authors might want to delete these results entirely, though it doesn't actually damage the manuscript to leave them in.

#188, delete "Moreover."

#190, I don't think that retaining the current-voltage relationship indicates that the channel properties are unchanged. Authors may have a specific more focused message in mind (regarding NR2A vs NR2B?). Perhaps the simplest way to handle it would be to change the wording to something like "suggesting little impact on channel properties."

#205, replace "while" with "though."

#229, there is a discrepancy between the text values and the illustration of Fig 4b. Please fix this.

#233, delete "Furthermore."

#262 the narrative structure "next" is heavily used in this manuscript. In this and many other manuscripts it is over-used (as is "then"). Here, I would write the sentence as "We used biochemical assays to determine...from mice."

#276, delete "further."

Authors add supplemental material to show development of RIM1. While certainly not the focus of the manuscript, they report in the supplemental legend that the markers increase with development, but in my copy, I see no developmental increase in GluN1.

#306 heading misuses the word "determine," perhaps inadvertently. I would replace it with a word like "affect" or "influence."

#311, replace "After that," with "After transduction."

#314-15 sentence should read ((GluN2B/NMDAR), since GluN2B..."

#338-9, sentence should read "GluA1 subunit, finding that RIM1..."

#342, delete "then."

#365, "...RIM1 level determines..." is a particularly notable misuse of the word "determines." Please modify.

#369-70, neither forskolin nor rolipram should be capitalized.

#382 "To explore this hypothesis" needs to be reworded or explained.

#387 etc, this reader finds the evidence that the mechanism is specifically and exclusively via the recycling pool unconvincing.

#397-8, I would shorten/reword as "Rab11 and Rab4 participate in...Piguel et al., 2014), but previous work..."

#417, delete the first part of the sentence; it should begin as "we used..."

The reasoning behind this section seems weak to me. A synopsis should include both pre-

and postsynaptic markers. If one sees only one, what does it mean? Is this sort of visual colocalization interpretable?

#441, replace "an" with "a."

DISCUSSION

The discussion seems wordy and in places repetitive. I'm not happy about authors citing many of their figures here (isn't that what the Results are for?) but this is for the Editor to decide.

#444-452, the two sentences seem to say virtually the same thing. Please delete one (and if necessary modify the other).

#460, replace "site" with "terminal."

#477 sentence should be shortened/rewritten, for example : "A number of active zone molecules...vesicle release, including SNARE components and complexin, have been..."

#488, please delete "Although it is....to recycled AMPARs."

#500-04, this sentence needs shortening and combining, thus (for example) "...of synaptic NMDARs, suggesting that RIM1...and GluN2B/NMDARs. GluN2B/NMDARs and GluN2A/NMDARs are equally affected..."

#507-8, I would write "...2007), suggesting that trafficking mechanisms may...." I would delete the entire remainder of the paragraph beginning "In addition," which seems both speculative and distracting/extraneous.

#513, should read "...have been found in both cortex and..."

#529 etc. needs shortening, as "perirhinal-dependent cognitive function, suggesting that postsynaptic RIM1....learning and memory. This is consistent with previous studies showing that...cognitive deficits, since the presynaptic roles of RIM1 do not seem sufficient to cause....(Powell et al., 2004)." and delete "this also..."

Reviewer #5 (Remarks to the Author):

I raised a few concerns regarding the pre-embedding immunogold labeling of the pre- and postsynapse for RIM1.

Most of my comments the authors answered sufficiently.

-The inclusion of two more animals is very good and strengthens the result of the immunogold experiments.

-Further, now the methods and how sampling for quantification is done are described much clearer and more detailed.

-In addition, the new gallery of pre-embedding results presented in the supplemental material is very useful and gives the reader the opportunity to see more than one image example.

-The sample size is absolutely sufficient

-However, regarding the background determination, I am not fully satisfied.

The contrast-enhanced images presented in the response are not convincing to me. To reliably determine a possible background staining, a negative control for pre-embedding

immunogold labeling would be required, ideally on the according KO. An antibody may react specifically in Western blotting or immunofluorescence, but this does not necessarily mean that in pre-embedding (or post, does not matter) immunogold labelings the antibody reacts comparable. Further, silver enhancement might cause problems, blocking might be less etc.

However, the manuscript improved overall a lot and due to the increased number of animals for the pre-embedding experiments and the newly presented images, I strongly support publication of the manuscript in Nature Communication.

Some minor comments:

Please indicate animal number also in the legend

Line 332: the surface

Line 466: Grabner et al. as a reference as well? Please check once more, if only Wang et al., 1997 is sufficient

Acknowledgement: please correct: Südhof

Fig. 5 please align panels d,e,f, g better

Fig. 8 the plots in f and g have different sizes

REVIEWERS' COMMENTS:

Reviewer #4 (Remarks to the Author):

This manuscript provides extensive new data suggesting a significant functional role for postsynaptic RIM1 in regulating NMDAR signaling in hippocampus. Results overall are self-consistent and reasonably convincing. My generally substantial enthusiasm for this work is somewhat reduced by several remaining concerns, some which can be easily addressed. The text should be improved in several places, especially by providing a shorter and more focused Discussion. In several places I think authors oversell their work: They provide good evidence for their core findings; in this reader's opinion making claims beyond their data only weakens their argument. I provide specific details and suggestions below. Authors were careful to blind parts of the study (electrophysiology and behavior); it's unfortunate that they apparently failed to blind other parts, but I don't consider that a fatal flaw.

Response

We thank the reviewer for the thoughtful and positive comments.

SPECIFICS/DETAILS:

I proceed in sequence through text, for my convenience and that of authors.

ABSTRACT

#30, replace "whereas" with "but."

#37 "determine" is inappropriately used here and several other places through the text. Perhaps authors don't quite understand the connotations of that word, but the unfortunate consequence is to considerably exaggerate the findings. Here, one could write (for example) "...RIM1 levels...influence both constitutive and ...

Response

Thanks for the suggestions. We have corrected them in the revised manuscript.

INTRODUCTION

#48 delete "the regulation of."

#51, should read "...that lead to rapid insertion of AMPARs into..."

#59, insert "may" between "or" and "be sorted."

#81/82, should read "...proteins, including SNAP25, ..."

#97, put a comma after "N-terminus."

#98-99, the last sentence is too strong. Perhaps authors are concerned that Nature Methods will review their manuscript; at this point they should instead try to provide a realistic and balanced view of their findings for the benefit of potential readers. I would say (for example) "...our results identify a substantial

role for postsynaptic...and suggest that this mechanism is important for ..."

Response

We appreciate the reviewer for these thoughtful suggestions. We have modified our manuscript accordingly.

RESULTS

#117, replace "that couples" with "which couples."

#126, I am no biochemist, but it seems that the RIM1 signal (e.g. 1b) is weak. Per se this does not detract from the main result, but if authors want to argue that a rather modest amount of postsynaptic RIM1 plays an important functional role (which is what I take to be their message) it might be best to acknowledge this explicitly from the beginning, and say (for example) "but also in the PSD fraction, though at lower levels."

#133, should read "...326 (79.5%) of presynaptic... and on 189 (46.1%) of postsynaptic..."

Response

Thanks for these suggestions. We have modified our manuscript accordingly.

The material shown in Supplemental figure 1 is ok, but is reproduced too dark; authors should adjust the tonal balance so that the silver-enhanced gold is

immediately obvious.

Response

Thanks for the valuable suggestion. We have slightly adjusted the tonal balance of Supplementary Figure 1 in the revised manuscript.

#146, I'm a little puzzled by the punctate colocalization work. Assessments of this type are technically difficult and should have been performed in a blinded manner, but I'm not sure what the goal was in any case. Is this really a satisfactory way to determine whether RIM1 is at the active zone or at the PSD? Perhaps authors have a different model in mind, in which RIM1 is in a cytoplasmic compartment? Authors might want to delete these results entirely, though it doesn't actually damage the manuscript to leave them in.

Response

We understand the reviewer's concern, and we agree with the reviewer that this colocalization analysis is not a satisfactory way to determine the synaptic localization of RIM1. Therefore, we deleted this result (the original Supplementary Figure 2b-d) in the revised manuscript to make it more focus.

#188, delete "Moreover."

#190, I don't think that retaining the current-voltage relationship indicates that

the channel properties are unchanged. Authors may have a specific more focused message in mind (regarding NR2A vs NR2B?). Perhaps the simplest way to handle it would be to change the wording to something like "suggesting little impact on channel properties."

#205, replace "while" with "though."

Response

Thanks for these suggestions. We have corrected them in the revised manuscript.

#229, there is a discrepancy between the text values and the illustration of Fig 4b.

Please fix this.

Response

We are very sorry for this mistake. The number for control group should be $50.7 \pm 1.4\%$, not $59.7 \pm 1.4\%$. We have corrected it in the revised manuscript.

#233, delete "Furthermore."

Response

Thanks for these suggestions. We have deleted it in the revised manuscript.

#262 the narrative structure "next" is heavily used in this manuscript. In this

and many other manuscripts it is over-used (as is "then"). Here, I would write the sentence as "We used biochemical assays to determine...from mice."

#276, delete "further."

Response

We appreciate the reviewer's suggestion. We have modified our manuscript accordingly.

Authors add supplemental material to show development of RIM1. While certainly not the focus of the manuscript, they report in the supplemental legend that the markers increase with development, but in my copy, I see no developmental increase in GluN1.

Response

We agree with the reviewer that developmental increase of GluN1 is not obvious. We modified our figure legend in the revised manuscript as *"Expression of RIM1 gradually increases during development, as well as that of GluN2A, and GluN2B."*

#306 heading misuses the word "determine," perhaps inadvertently. I would replace it with a word like "affect" or "influence."

#311, replace "After that," with "After transduction."

#314-15 sentence should read ((GluN2B/NMDAR), since GluN2B...."

#338-9, sentence should read "GluA1 subunit, finding that RIM1..."

#342, delete "then."

#365, "...RIM1 level determines..." is a particularly notable misuse of the word "determines." Please modify.

#369-70, neither forskolin nor rolipram should be capitalized.

#382 "To explore this hypothesis" needs to be reworded or explained.

Response

Thanks for these valuable suggestions. We have re-worded them in the revised manuscript.

#387 etc, this reader finds the evidence that the mechanism is specifically and exclusively via the recycling pool unconvincing.

Response

We appreciate the reviewer's suggestions. We have modified it as "*These data suggest that the decrease of surface-localized NMDARs after RIM1 KD is probably due to impaired receptor recycling.*" in the revised manuscript.

#397-8, I would shorten/reword as "Rab11 and Rab4 participate in...Piguel et al., 2014), but previous work..."

#417, delete the first part of the sentence; it should begin as "we used..."

Response

Thanks for these thoughtful suggestions. We have corrected them in the revised manuscript.

The reasoning behind this section seems weak to me. A synapse should include both pre- and postsynaptic markers. If one sees only one, what does it mean? Is this sort of visual colocalization interpretable?

Response

We understand the reviewer's concern. It is better to use both pre- and postsynaptic markers to indicate a synapse. Sometimes, it is difficult to label more than two proteins at the same time, mainly due to the limitation of antibodies. Therefore, we use only one postsynaptic marker to label the synapses.

#441, replace "an" with "a."

Response

Sorry for this mistake. We have corrected it in the revised manuscript.

DISCUSSION

The discussion seems wordy and in places repetitive. I'm not happy about

**authors citing many of their figures here (isn't that what the Results are for?)
but this is for the Editor to decide.**

Response

We appreciate the reviewer's suggestion. We have re-polished the discussion section in the revised manuscript and deleted the unnecessary description.

#444-452, the two sentences seem to say virtually the same thing. Please delete one (and if necessary modify the other).

#460, replace "site" with "terminal."

#477 sentence should be shortened/rewritten, for example : "A number of active zone molecules....vesicle release, including SNARE components and complexin, have been..."

#488, please delete "Although it is....to recycled AMPARs."

#500-04, this sentence needs shortening and combining, thus (for example) "...of synaptic NMDARs, suggesting that RIM1...and GluN2B/NMDARs. GluN2B/NMDARs and GluN2A/NMDARs are equally affected..."

#507-8, I would write "...2007), suggesting that trafficking mechanisms may...." I would delete the entire remainder of the paragraph beginning "In addition," which seems both speculative and distracting/extraneous.

#513, should read "...have been found in both cortex and..."

#529 etc. needs shortening, as "perirhinal-dependent cognitive function,

suggesting that postsynaptic RIM1.....learning and memory. This is consistent with previous studies showing that...cognitive deficits, since the presynaptic roles of RIM1 do not seem sufficient to cause....(Powell et al., 2004)." and delete "this also..."

Response

We sincerely appreciate the reviewer's valuable suggestions. We have modified our manuscript accordingly.

Reviewer #5 (Remarks to the Author):

I raised a few concerns regarding the pre-embedding immunogold labeling of the pre- and postsynapse for RIM1.

Most of my comments the authors answered sufficiently.

-The inclusion of two more animals is very good and strengthens the result of the immunogold experiments.

-Further, now the methods and how sampling for quantification is done are described much clearer and more detailed.

-In addition, the new gallery of pre-embedding results presented in the supplemental material is very useful and gives the reader the opportunity to see more than one image example.

-The sample size is absolutely sufficient

We appreciate that the reviewer thinks we have satisfactorily addressed his concern.

-However, regarding the background determination, I am not fully satisfied.

The contrast-enhanced images presented in the response are not convincing to me. To reliably determine a possible background staining, a negative control for pre-embedding immunogold labeling would be required, ideally on the according KO. An antibody may react specifically in Western blotting or immunofluorescence, but this does not necessarily mean that in pre-embedding (or post, does not matter) immunogold labelings the antibody reacts comparable. Further, silver enhancement might cause problems, blocking might be less etc.

Response

We agree with the reviewer that negative control for pre-embedding immunogold labeling would be required. We will keep this in mind and will design more reliable experiment as negative control in future. Thanks again for your consideration.

However, the manuscript improved overall a lot and due to the increased number of animals for the pre-embedding experiments and the newly presented images, I strongly support publication of the manuscript in Nature

Communication.

We appreciate that the reviewer thinks our manuscript is suitable for publication in Nature Communication.

Some minor comments:

Please indicate animal number also in the legend

Response

Thanks for the suggestion. We have added it in the Figure legend of the revised manuscript.

Line 332: the surface

Response

Thanks for the suggestion. We have added “the” in the revised manuscript.

Line 466: Grabner et al. as a reference as well? Please check once more, if only

Wang et al., 1997 is sufficient

Response

Thanks for the suggestion. We have added this reference in our revised manuscript.

Acknowledgement: please correct: Südhof

Response

Sorry for this mistake. We have corrected it in the revised manuscript.

Fig. 5 please align panels d,e,f, g better

Response

Thanks for the suggestion. We have aligned these panels in the revised manuscript.

Fig. 8 the plots in f and g have different sizes

Response

Sorry for the mistake. We have corrected it in the revised manuscript.